# INDUCTIVE GRADIENT ADJUSTMENT FOR SPECTRAL BIAS IN IMPLICIT NEURAL REPRESENTATIONS

## ABSTRACT

Implicit Neural Representations (INRs) as a versatile representation paradigm have achieved success in various computer vision tasks. Due to the spectral bias of the vanilla multi-layer perceptrons (MLPs), existing methods focus on designing MLPs with sophisticated architectures or repurposing existing training techniques for highly accurate INRs. In this paper, we delve into the linear dynamics model of MLPs and theoretically identify the empirical Neural Tangent Kernel (eNTK) matrix as a reliable link between spectral bias and training dynamics. Based on eNTK matrix, we propose a practical inductive gradient adjustment method, which could purposefully improve the spectral bias via inductive generalization of eNTK-based gradient transformation matrix. We evaluate our method on different INRs tasks with various INR architectures and compare to existing training techniques. The superiority representation performance clearly validate the advantage of our proposed method. Armed with our gradient adjustment method, better INRs with more enhanced texture details and sharpened edges can be learned from the training data by tailored improvements on spectral bias.

## 1 INTRODUCTION

The main idea of implicit neural representations (INRs) is using neural networks such as multi-layer perceptrons (MLPs) to parameterize discrete signals in an implicit and continuous manner. Benefiting from the continuity and implicit nature, INRs have gained great attention as a versatile signal representation paradigm and achieved state-of-the-art performance across a wide range of computer vision tasks such as signal representation (Sitzmann et al., 2020; Liang et al., 2022; Saragadam et al., 2023; Guo et al., 2023), 3D shape reconstruction (Cai et al., 2024; Shi et al., 2024a; Zhu et al., 2024) and novel view synthesis (Mildenhall et al., 2021; Müller et al., 2022; Fathony et al., 2020).

However, obtaining high-precision INRs is non-trivial, MLPs with ReLU activation function often fails to represent high-frequency details. Such tendency of MLPs to represent simple patterns of target function is referred to spectral bias (Rahaman et al., 2019; Xu, 2018). To improve the performance of INRs, great efforts have been made to alleviate or overcome this bias of MLPs. These efforts have primarily focused on how MLPs are constructed such as complex input embeddings (Tancik et al., 2020; Xie et al., 2023; Müller et al., 2022) and sophisticated activation functions (Sitzmann et al., 2020; Saragadam et al., 2023). Recently, two studies find that training modifications based on existing training techniques, i.e., Fourier reparameterized training (FR) (Shi et al., 2024a) and batch normalization (BN) (Cai et al., 2024), can improve training dynamics, thereby overcoming the spectral bias without altering inference structures.

Despite these works serendipitously discover that such FR and BN training could alleviate the spectral bias and lead to better representation performance, the mechanism behind their improvement is still unclear. Moreover, there is no clear guidance on the choice of Fourier bases matrix for reparameterization or batch normalization layer to confront spectral bias with varying degrees in a variety of INR tasks. Lack of reliable insight and guidance results in sub-optimal improvements in their broader applications. Therefore, how to adjust training dynamics of MLPs to purposefully overcome spectral bias remains an open and valuable research challenge.

In this paper, we delve into the linear dynamics model of MLPs and propose an effective training dynamics adjustment strategy guided by theoretical derivation to purposefully alleviate the spectral

bias issue. Our strategy allows for tailored impacts on spectral bias based on the spectrum of target signals, thereby achieving more precise INRs. Specifically, these impacts on spectral bias are tailored by purposefully adjusting the spectrum of the Neural Tangent Kernel (NTK) matrix (Jacot et al., 2018), which closely connects the spectral bias with training dynamics. Given that NTK matrix is not available in most cases, we theoretically identify the empirical NTK matrix as a tractable surrogate. Empirical results corroborate this theoretical results. Based on eNTK matrix, we further propose an inductive gradient adjustment (IGA) method which could purposefully improve the spectral bias via inductive generalization of eNTK-based gradient transformation matrix with millions of data points. We give both theoretical and empirical analysis of our method, demonstrating how it tailors impacts on spectral bias. Besides, our IGA method can work with previous structural improvement methods such as positional encoding (Tancik et al., 2020) and periodic activation function (Sitzmann et al., 2020). We validate our method in various vision applications of INRs and compare it to previous training dynamics adjustment methods. Experimental results shows the superiority of training dynamics adjustment strategy. Our contributions are summarized as follows:

- We connect spectral bias with linear dynamics model of MLPs and derive an training dynamics adjustment strategy guided by a theoretical standpoint to purposefully improve the spectral bias, i.e., tailored impacts on spectral bias based on the spectrum of target signals.

- We propose a practical inductive gradient adjustment method which could purposefully mitigate spectral bias of MLPs even with millions of data points. Theoretical and empirical analyses show how our method tailors impacts on spectral bias.

- We provide detailed experimental analyses across a wide range of implicit neural representation tasks. Comparing to previous training dynamics methods, our method allows for tailored improvement on spectral bias of commonly MLPs and provides implicit neural representations with more high-frequency details.

## 2 RELATED WORK

**Implicit neural representations.** Recently, implicit neural representations (INRs) representing the discrete signal as an implicit continuous function by MLPs have gained lots of attention. Comparing to traditional grid-based signal representation method, INRs have shown the remarkable representation accuracy and memory-efficient property in a wide range of representation tasks such as 1D audio representation (Sitzmann et al., 2020; Kim et al., 2022), 2D image representation (Klocek et al., 2019; Strümpler et al., 2022) , 3D shape representation (Park et al., 2019; Martel et al., 2021), novel view synthesis (Mildenhall et al., 2021; Saragadam et al., 2023) and virtual reality (Deng et al., 2022). However, vanilla MLPs with ReLU activation function fails to represent complex signals. Therefore, various modifications have been studied. One category modifications focus on the embedings of inputs. Mildenhall et al. (2021) find that using positional encoding (PE) as inputs enhances the representation of high-frequency details in neural radiance fields. Further, Takikawa et al. (2021); Martel et al. (2021); Xie et al. (2023) adopt learned features to encode inputs, achieving better INRs. A different category of modifications concentrates on activation functions. Sitzmann et al. (2020) found that periodic functions, such as Sine function, can obtain more accurate representation than ReLU activation function. Sine functions are also explored by Fathony et al. (2020) in multiplicative filter networks. Saragadam et al. (2023) adopt a complex Gabor wavelet activation and achieve robust and accurate representations. Considering a broader framework, the aforementioned work can be summarized as efforts to construct implicit neural representation models or MLPs with more powerful representational capacity (Yüce et al., 2022). Recently, another category, distinct from those previously discussed, focuses on the training process of MLPs. Shi et al. (2024a) find that learning parameters in the Fourier domain, i.e., Fourier reparameterized training (FR), can improve approximation accuracy of INRs without altering the inference structure. Cai et al. (2024) repurpose that the classic batch normalization layer (BN) can also improve performance of INRs.

**Spectral bias.** Spectral bias typically refers to a learning bias of MLPs that MLPs tend to learn simple patterns of training data or the low-frequency components of the target function. Great efforts have been made to dissect this bias. Rahaman et al. (2019) find that spectral bias varies with the width and depth of MLPs and the complexity of input manifold by experimental and theoretical analyses. Generally, wider or deeper MLPs can learn high frequency information more efficiently, and increasing the complexity of input manifolds serves a similar purpose. Xu (2018) attribute this

bias of MLPs with Tanh activation function to the uneven distribution of gradients in the frequency domain through Fourier decomposition. From the perspective of linear dynamics of MLPs, spectral bias is induced by the uneven distributed eigenvalues of the corresponding Neural Tangent Kernel (NTK) matrix (Jacot et al., 2018; Arora et al., 2019; Ronen et al., 2019). Inspired by this theoretical result, Tancik et al. (2020); Shi et al. (2024a); Cai et al. (2024) try to elucidate spectral bias of different models by observing eigenvalue distribution and find that PE, FR and BN help to attain a more uniform eigenvalue distribution of the NTK matrix. Similarly, Geifman et al. (2023) theoretically discuss impacts of modifying the NTK matrix spectrum in infinitely wide networks.

In this paper, we adopts the insight of attaining more uniform eigenvalue distribution of the NTK matrix to overcome the spectral bias. For the first time, we show that modifying the eNTK matrix spectrum can have the similar impacts on spectral bias and propose a practical inductive gradient adjustment method for improving the spectral bias of INRs.

## 3 METHOD

In this section, we firstly review the linear dynamics model of MLPs, indicating the potential of adjusting the spectrum of NTK matrix to purposefully improve the spectral bias. Then we analyze the NTK-based adjustment method and show that the intractability of the NTK matrix renders it impractical. By further theoretical analysis, we identify empirical NTK matrix as a tractable proxy, but it succumbs to dimensionality challenges as the data size increases. Therefore, we propose a practical gradient adjustment method, which could purposefully improve the spectral bias via inductive generalization of eNTK-based gradient adjustments.

### 3.1 BACKGROUND: CONNECTING SPECTRAL BIAS WITH TRAINING DYNAMICS

Considering a discrete signal $S = \{(\mathbf{x}_i, y_i)\}_{i=1}^N$ over $\mathbb{R}^{d_0} \times \mathbb{R}^1$, the INR of this signal denoted as $f(\boldsymbol{x}, \Theta)$. An MLP with constant width $m$ is adopted to parameterize $f(\boldsymbol{x}, \Theta)$. Parameters $\Theta$ are optimized via gradient descent under the supervision of the squared error loss $\mathcal{L}(\mathbf{x}; \Theta) = \frac{1}{2} \sum_{i=1}^N (f(\mathbf{x}_i; \Theta) - y_i)^2$. We denote the residual $(f(\boldsymbol{x}_i; \Theta_t) - y_i)_{i=1}^n$ at time step $t$ as $\boldsymbol{r}_t$. Training dynamics of $\boldsymbol{r}_t$ can be approximated by the linear dynamics model when the width $m$ is large enough and learning rate $\eta$ is small enough (Du et al., 2018; Arora et al., 2019; Lee et al., 2019; Geifman et al., 2023): $\boldsymbol{r}_t = (\boldsymbol{I} - \eta\boldsymbol{K})\boldsymbol{r}_{t-1}$, where $\boldsymbol{K}$ is the NTK matrix on $S$ defined by $\boldsymbol{K} = \mathbb{E}_{\Theta_0}[\nabla_\Theta f(\boldsymbol{X}; \Theta_0)^\top \nabla_\Theta f(\boldsymbol{X}; \Theta_0)]$. With the eigenvalue decomposition of $\boldsymbol{K} = \sum_{i=1}^N \lambda_i \boldsymbol{v}_i \boldsymbol{v}_i^\top$, training dynamics of $\boldsymbol{r}_t$ can be characterized as follows (Arora et al., 2019):

$$||\mathbf{r}_t||_2 = \sqrt{\sum_{i=1}^N (1 - \eta\lambda_i)^{2t} (\boldsymbol{v}_i^\top \boldsymbol{y})^2}. \tag{1}$$

Equation 1 shows that convergence rate of $f(\boldsymbol{x}, \Theta)$ at the projection direction $\boldsymbol{v}_i^\top$ with larger $\lambda_i$ will be faster. For vanilla MLPs, projection directions related to high frequencies are consistently assigned to small eigenvalues, while those related to low frequencies correspond to larger eigenvalues (Ronen et al., 2019; Bietti & Mairal, 2019; Heckel & Soltanolkotabi, 2020). The uneven spectrum of $\boldsymbol{K}$ lead to extremely slow convergence to the high frequency components of signal thereby leading to lower Peak Signal-to-Noise Ratio (PSNR) values in INRs. Following this insight, recent works (Tancik et al., 2020; Bai et al., 2023; Shi et al., 2024a; Cai et al., 2024) suggest that MLPs with balanced eigenvalues of $\boldsymbol{K}$ are less affected by spectral bias and achieve better PSNR values.

### 3.2 IMPROVING SPECTRAL BIAS VIA TRAINING DYNAMICS SHARING

As previously discussed, eigenvalues of $\boldsymbol{K}$ almost governs convergence rates of MLPs to different components, and uneven spectrum of $\boldsymbol{K}$ lead to slow convergence of high frequency components. It is natural to adjust the spectrum of $\boldsymbol{K}$ purposefully to tailor the improvement of spectral bias. Following this intuition, we focus on the adjustment of eigenvalues of $\boldsymbol{K}$. This can be naturally achieved by constructing a transformation matrix $\boldsymbol{S}$ that has the same eigenvectors as $\boldsymbol{K}$ as follows:

$$\Theta_{t+1} = \Theta_t - \eta\nabla_\Theta f(\boldsymbol{X}; \Theta)\boldsymbol{S}\boldsymbol{r}_t, \tag{2}$$

where $\boldsymbol{S} = \sum_{i=1}^N (g_i(\lambda_i)/\lambda_i)\boldsymbol{v}_i\boldsymbol{v}_i^\top$; $\{g_i(\lambda_i)\}_{i=1}^N$ denotes the desired spectrum. Geifman et al. (2023) theoretically analyze spectral impacts of equation 2 on $\boldsymbol{K}$ and validate it using a toy MLP with limited synthetic data.

However, equation 2 is practically infeasible in a broad spectrum of INRs tasks. The first challenge is that an analytical expression for $\boldsymbol{K}$ is difficult to derive (Xu et al., 2021; Novak et al., 2022; Wang et al., 2023; Shi et al., 2024b). Specifically, the analytical expression for $\mathbb{E}_{f \sim \mathcal{N}(0,\boldsymbol{\Sigma})}[\dot{\sigma}(f(x))\dot{\sigma}(f(x'))]$, which is the key of $\boldsymbol{K}$, becomes intractable as the depth of the network increases. The second challenge is that the size of matrix $\boldsymbol{K}$ grows quadratically with the increase of the number of data points (Mohamadi et al., 2023). Considering Kodak image fitting task, it requires to store and decompose a matrix containing about 1 trillion entries, which takes over eight terabytes in memory if stored in double precision.

To overcome the first challenge, we try to adjust eigenvalues of empirical NTK (eNTK) matrix $\tilde{\boldsymbol{K}}$ as $\tilde{\boldsymbol{K}}$ inherently exists in the linear dynamics model of MLPs with arbitrary width. Despite numerous studies (Xu et al., 2021; Novak et al., 2022; Mohamadi et al., 2023) exploring the connection between $\tilde{\boldsymbol{K}}$ and $\boldsymbol{K}$, impacts on spectral bias by $\tilde{\boldsymbol{K}}$-based adjustment are not evidently equivalent to $\boldsymbol{K}$-based adjustment. Through our theoretical analysis in Theorem 4.1, we prove that projection directions of $\tilde{\boldsymbol{K}}$ as well as eigenvalues converge to the corresponding parts of $\boldsymbol{K}$ as the network width increases. By this theoretical result, impacts on spectral bias by $\tilde{\boldsymbol{K}}$-based adjustment can be approximately equivalent to that of $\boldsymbol{K}$-based adjustment. We construct the transformation matrix $\tilde{\boldsymbol{S}}$ based on $\tilde{\boldsymbol{K}}$ and adjust the gradients as follows:

$$\Theta_{t+1} = \Theta_t - \eta \nabla_{\Theta_t} f(\boldsymbol{X}; \Theta_t) \tilde{\boldsymbol{S}} \boldsymbol{r}_t, \tag{3}$$

where $\tilde{\boldsymbol{S}} = \sum_{i=1}^{N}(g_i(\tilde{\lambda}_i)/\tilde{\lambda}_i)\tilde{\boldsymbol{v}}_i\tilde{\boldsymbol{v}}_i^\top$; $\tilde{\boldsymbol{K}} = \nabla_{\Theta_t} f(\boldsymbol{X}; \Theta_t)^\top \nabla_{\Theta_t} f(\boldsymbol{X}; \Theta_t) = \sum_{i=1}^{N} \tilde{\lambda}_i \tilde{\boldsymbol{v}}_i \tilde{\boldsymbol{v}}_i^\top$. Thus, $\tilde{\boldsymbol{K}}$ not only links spectral bias with network training dynamics but also serves as a tractable estimate.

Although $\tilde{\boldsymbol{K}}$ avoids intractability, it still encounters the curse of dimensionality as $N$ increases. In Theorem 4.2, we have that training dynamics of $\boldsymbol{r}_t$ could be estimated via inductive generalization of the linear dynamics model of sampled data points and analyzable error. Inspired by this theoretical foundation, we propose a practical inductive gradient adjustment method. Specifically, assumed that $\{(\mathbf{x}_i, y_i)\}_{i=1}^{N}$ have been sorted based on the proximity of input samples, we divide these samples into $n$ groups, where each group is $\boldsymbol{X}^j = \{(\boldsymbol{x}_i^j, y_i^j)\}_{i=1}^{p}$. Then we sample one point from each group to form the sample set $\boldsymbol{X}_e$. We denote the eNTK matrix on $\boldsymbol{X}_e$ as the empirical inductive NTK matrix $\tilde{\boldsymbol{K}}_e$ and construct the corresponding transformation matrix $\tilde{\boldsymbol{S}}_e$. Then inductive adjustments are generalized to gradients of whole data points as follows:

$$\Theta_{t+1} = \Theta_t - \eta \sum_{i=1}^{p} \nabla_{\Theta_t} f(\boldsymbol{X}_i, \Theta_t) \tilde{\boldsymbol{S}}_e \boldsymbol{r}_t^i, \tag{4}$$

where $\boldsymbol{X}_i = \{(\boldsymbol{x}_i^j, y_i^j)\}_{j=1}^{n}$ and $\boldsymbol{r}_t^i$ denotes the corresponding residual vector at time step $t$; $\tilde{\boldsymbol{S}}_e = \sum_{i=1}^{N}(g_i(\tilde{\tilde{\lambda}}_i)/\tilde{\tilde{\lambda}}_i)\tilde{\boldsymbol{v}}_i\tilde{\boldsymbol{v}}_i^\top$; $\tilde{\boldsymbol{K}}_e = \nabla_{\Theta_t} f(\boldsymbol{X}_e; \Theta_t)^\top \nabla_{\Theta_t} f(\boldsymbol{X}_e; \Theta_t) = \sum_{i=1}^{N} \tilde{\tilde{\lambda}}_i \tilde{\boldsymbol{v}}_i \tilde{\boldsymbol{v}}_i^\top$. As shown in equation 4, all $\boldsymbol{r}_i^t$ are linearly transformed by $\boldsymbol{S}_{tds}^t$. Please note that for each vector $\boldsymbol{r}_i^t$ (where $i = 1, \ldots, p$), its $j$-th element is scaled by the $j$-th column of the matrix $\tilde{\boldsymbol{S}}_e$. And the $j$-th element of these vectors $\boldsymbol{r}_i^t$ all belong to the shared interval $\boldsymbol{X}_j$. Therefore, our inductive gradient adjustments are generalized to other data points in each group $\boldsymbol{X}^j$. From our Theorem 4.2, although the inductive generalization by $\tilde{\boldsymbol{K}}$ introduces errors; these errors can be reduced by increasing data similarity, and wider MLPs also provide a chance to reduce these errors. Further, in most INR tasks, our experimental results illustrate that these errors could be neglected.

## 3.3 Implementation Details

**Construction Strategy.** As we have introduced in equation 2, we construct the transformation matrix by the desired spectrum. We have that $\tilde{\boldsymbol{K}}_e = \sum_{i=1}^{m} \lambda_i \boldsymbol{v}_i \boldsymbol{v}_i^\top$ and assume that $\lambda_1 > \cdots > \lambda_m > 0$. To purposefully improve the spectral bias, we balance the eigenvalues of different eigenvectors; impacts are tailored by managing the number of balanced eigenvalues. For vanilla gradient descent, we have $\tilde{\boldsymbol{S}}$ as follows:

$$\tilde{\boldsymbol{S}}_e = \sum_{i=start}^{end} \frac{\lambda_{start}}{\lambda_i} \mathbf{v}_i \mathbf{v}_i^\top + \sum_{i \notin [start, end]} \lambda_i \mathbf{v}_i \mathbf{v}_i^\top. \tag{5}$$

Generally, $start$ is fixed at 1; $end$ represents the controlled spectral range. The larger the value of $end$, the stronger impacts over spectral bias.

Due to the introduction of adaptive learning rates and momentum in Adam, training dynamics of MLPs becomes exceptionally challenging and remains an open problem. Intuitively, momentum is the linear combination of previous gradients, which implies that momentum has the similar direction with current adjusted gradients. This inspires us to extends the equation 4 to Adam. Adaptive learning rates typically result in larger update steps for parameters (Kingma, 2014; Wilson et al., 2017; Reddi et al., 2019). Therefore, we utilize $\lambda_{end}$ to balance eigenvalues in equation 5 for better convergence. Experiments show its promising performance.

**Sampling Strategy.** As illustrated in Sec. 3.2, we need to sample data points from groups to compute $\tilde{\boldsymbol{K}}$. Discussing all sampling strategies is beyond the scope of our work. For coordinate inputs, We partition inputs by Euclidean distance of samples into $n$ groups with $p$ points and largest residual points in each group are sampled. Details of hyperparameters $n$ and $p$ for different settings will be introduced in the experimental section. We provide ablation experiments in our Appendix B to analyze the effect of group size $p$ and points.

**Multidimensional output approximation.** In practical INRs tasks, the output of MLPs is multidimensional in certain scenarios, such as color images. Comparing to the dimension of data points, it is trivial but still affects the efficiency. Therefore, we simply compute the Jacobian matrix by sum-of-logits which has been proved efficient in (Mohamadi et al., 2023; Shi et al., 2024b).

# 4 THEORETICAL ANALYSIS

In this section, we firstly introduce our analysis framework, referring to previous works (Du et al., 2018; Arora et al., 2019; Lee et al., 2019; Geifman et al., 2023). Building on this, we prove that adjustments based on eNTK matrix asymptotically converge to those based on NTK matrix as the network width increases. Then we delve into the estimate of training dynamics via sharing the linear dynamics model of few data points and give a theoretical analysis to its potential error.

**Analysis Framework.** For better analysis of NTK matrix, we refer previous works (Du et al., 2018; Arora et al., 2019; Lee et al., 2019; Geifman et al., 2023) and perform a theoretical analysis on the basis of the following settings: assuming that the training set $\{\boldsymbol{x}_i, y_i\}_{i=1}^{N}$ is contained in some compact set, a two-layer network $f(\boldsymbol{x}; \Theta)$ with $m$ neurons is formalized as follows to fit these data:

$$f(\boldsymbol{x}; \Theta) = (1/\sqrt{m}) \sum_{r=1}^{m} a_r \sigma(\boldsymbol{w}_r^\top \boldsymbol{x} + b_r), \tag{6}$$

where the activation function $\sigma$ satisfies that $|\sigma(0)|, \|\sigma'\|_\infty, sup_{x \neq x'}|\sigma'(x) - \sigma'(x')|/|x - x'| < \infty$. The detailed initialization scheme can be found in our Appendix A.1. The loss function is $\frac{1}{2} \sum_{i=1}^{N} (f(\boldsymbol{x}_i, \Theta_t) - y_i)^2$. We assume that $\boldsymbol{K}, \tilde{\boldsymbol{K}}$ are full rank indicating that they are positive definite. This assumption generally holds due to the complexity of neural networks. Then we have the following standard orthogonal spectral decomposition: $\boldsymbol{K} = \sum_{i=1}^{N} \lambda_i \boldsymbol{v}_i \boldsymbol{v}_i^\top$ and $\tilde{\boldsymbol{K}} = \sum_{i=1}^{N} \tilde{\lambda}_i \tilde{\boldsymbol{v}}_i \tilde{\boldsymbol{v}}_i^\top$, which eigenvalues are indexed in descending order of magnitude. We construct the corresponding transformation matrices as $\boldsymbol{S} = \sum_{i=1}^{N} (g_i(\lambda_i)/\lambda_i) \boldsymbol{v}_i \boldsymbol{v}_i^\top$ and $\tilde{\boldsymbol{S}} = \sum_{i=1}^{N} (g_i(\tilde{\lambda}_i)/\tilde{\lambda}_i) \tilde{\boldsymbol{v}}_i \tilde{\boldsymbol{v}}_i^\top$. Although $\lim_{m \to \infty} \|\boldsymbol{K} - \tilde{\boldsymbol{K}}\|_F = 0$ (Jacot et al., 2018; Lee et al., 2019; Geifman et al., 2023), it is non-trivial to show that $\boldsymbol{S}, \tilde{\boldsymbol{S}}$ have the similar impacts on the spectrum. Thanks to the well-established theory of matrix perturbation analysis (Yu et al., 2015; Davis & Kahan, 1970; Baumgärtel, 1984), we have the following Theorem 4.1.

**Theorem 4.1.** *The following standard orthogonal spectral decomposition exists:* $\boldsymbol{K} = \sum_{i=1}^{N} \lambda_i \boldsymbol{v}_i \boldsymbol{v}_i^\top$ *and* $\tilde{\boldsymbol{K}} = \sum_{i=1}^{N} \tilde{\lambda}_i \tilde{\boldsymbol{v}}_i \tilde{\boldsymbol{v}}_i^\top$, *satisfying that* $\boldsymbol{v}_i^\top \tilde{\boldsymbol{v}}_i > 0$ *for* $i \in [N]$. *We denoted* $min\{\lambda_i - \lambda_{i+1}, \lambda_{i+1} - \lambda_{i+2}\}$ *as* $G$. *Let* $\{g_i(x)\}_{i=1}^{N}$ *be a set of Lipschitz continuous functions, with the Supremum of their Lipschitz constants denoted by* $k$, *with* $\eta < min\{((max(g(\lambda)) + min(g(\lambda)))^{-1}, (max(g(\tilde{\lambda})) + min(g(\tilde{\lambda})))^{-1}\}$, *for* $\epsilon > 0$, *there always exists* $M > 0$, *such that* $m > M$, *for* $i \in [N]$, *we have that* $|g(\lambda_i) - g(\tilde{\lambda}_i)| < \epsilon_1, \|\boldsymbol{v}_i - \tilde{\boldsymbol{v}}_i\| < \epsilon_2$; *furthermore, we have that:*

$$|(1 - \eta g(\tilde{\lambda}_i))^2 (\tilde{\boldsymbol{v}}_i^\top \boldsymbol{r}_t)^2 - (1 - \eta g(\lambda_i))^2 (\boldsymbol{v}_i^\top \boldsymbol{r}_t)^2| < \epsilon_3, \|\boldsymbol{r}_{t+1} - \tilde{\boldsymbol{r}}_{t+1}\| < \epsilon_4,$$

*where* $\epsilon_1 = k\epsilon$; $\epsilon_2 = (2^{3/2}\epsilon)/G$; $\epsilon_3 = \frac{8R_0}{G^2}(k\eta\epsilon^3 + (v+1)\epsilon^2 + (v^2 + Gv)\epsilon)$; $\epsilon_4 = \epsilon N((\frac{16vR_0}{G} + \eta kv^2 R_0) + \epsilon^2) + 2\epsilon$; $v, k, R_0$ *are constants.*

The detailed proof of Theorem 4.1 can be found in our Appendix A.3. Theorem 4.1 shows that eNTK-based gradient adjustment ($\tilde{S}$) exhibits almost the same level of control as NTK-based adjustment ($S$) across different feature directions as width increases, thereby leading to almost the same training dynamics of $r_{t+1}$ and $\tilde{r}_{t+1}$. More fundamentally, eigenvectors of $\tilde{K}$ converge one-to-one to eigenvectors of $K$, thus ensuring that $\tilde{S}$ impact the similar convergence directions as $S$. In Theorem 4.2, we show that training dynamics could be estimated by inductive generalization of the linear dynamics model of sampled data points.

**Theorem 4.2.** $X = \{x_i\}_{i=1}^N$ are partitioned by order into $n$ shared groups, where each group is $X_j = \{x_1^j, \ldots, x_p^j\}$ and $N = np$. For each group, one data point is sampled denoted as $x^j$ and these $n$ data points form $X_e = \{x^j\}_{j=1}^n, n \ll N$. There exists $\epsilon > 0$, for $j = 1, \ldots, n$ and any $1 \le i_1, i_2 \le p$, such that $|(f(x_{i_1}^j, \Theta_t) - y_{i_1}^j) - (f(x_{i_2}^j, \Theta_t) - y_{i_2}^j)|, \|\nabla_{\Theta_t} f(x_{i_1}^j) - \nabla_{\Theta_t} f(x_{i_2}^j)\| < \epsilon$. Then, for any $x_i$, assumed that $x_i$ belongs to $X_{j_i}$, such that:

$$|\Delta f(x_i, \Theta_t) - \nabla_{\Theta_t} f(x^{j_i})^\top [p \sum_{j=1}^n \nabla_{\Theta_t} f(x^j)(f(x^j) - y^j)]| < \epsilon(\frac{(\kappa + n^{3/2})R_0 + \kappa^2}{\sqrt{m}}) + \frac{\eta \kappa^3 R_0^2}{m^{3/2}},$$

where $\Delta f(x_i, \Theta_t) = (r_{t+1} - r_t)_i$ denotes the dynamics of $f(x_i, \Theta)$ at time step $t$; $R_0, \kappa$ are constants; $\nabla_{\Theta_t} f(x^{j_i}, \Theta)^\top \nabla_{\Theta_t} f(x^j)$ is the entry in $\tilde{K}_e(i, j)$.

The detailed proof can be found in our Appendix A.4. Theorem 4.2 offers us a new perspective that training dynamics can be estimated by inductive generalization of the linear dynamics model of sampled data points. Moreover, this error decreases as the similarity of data increases, and increasing $m$ provides a chance to reduce this error. Inspired by this, we construct the transformation matrix $\tilde{S}_e$ based on $\tilde{K}_e$ to purposefully improve the spectral bias. Practically, our empirical results show that this error is trivial to purposefully improve the spectral bias.

## 5 EMPIRICAL ANALYSIS ON SIMPLE FUNCTION APPROXIMATION

In this experiment, we firstly give an empirical validation to our Theorem 4.1 that impacts on spectral bias guided by $K$ and $\tilde{K}$ are similar. Then, we demonstrate that our inductive gradient adjustment method can purposefully improve the spectral bias of MLPs by managing the number of balanced eigenvalues of $\tilde{K}_e$.

To better analyze, we follow previous works (Xu, 2018; Shi et al., 2024a) and compute the relative error $\Delta_k$ between target signal $g$ and outputs $f_\Theta$ at frequency $k$ to show spectral bias:

$$\Delta_k = \frac{|\mathcal{F}_D[g](k) - \mathcal{F}_D[f_\Theta](k)|}{|\mathcal{F}_D[g](k)|}, \tag{7}$$

where $\mathcal{F}_D$ denotes the discrete Fourier transform.

**Experiment 1.** In this experiment, we aim to corroborate our theoretical results from the previous section, specifically: impacts of $\tilde{K}$-based gradient adjustment on spectral bias are similar to that of $K$-based gradient adjustment, and the differences decrease as the network width increases; inductive generalization via $\tilde{K}_e$ for gradient adjustment is effective, and increasing width helps reduce the estimate error. Therefore, the accurate computation of $K$ is the key for validation. Given that the accurate computation of $K$ is non-trivial, we adopt a two-layer MLP with a fixed last layer in Arora et al. (2019); Ronen et al. (2019), whose $K_{ij}$ can be computed by the formula $\frac{1}{4\pi}(x_i^\top x_j + 1)(\pi - arccos(x_i^\top x_j))$. This architecture facilitates theoretical analysis but has limited representational capacity for representing complex functions. To better demonstrate the spectral bias of MLPs, we constructed the following simple function $f : \mathbb{S}^1 \to \mathbb{R}^1$ that has been widely adopted by previous works (Ronen et al., 2019; Geifman et al., 2023):

$$f(\theta) = f(cos(\theta), sin(\theta)) = sin(0.4\pi\theta) + sin(0.8\pi\theta) + sin(1.6\pi\theta) + sin(3.2\pi\theta). \tag{8}$$

Then the two-layer MLP in Arora et al. (2019) is employed to parameterize this function with $N = 1024$ input samples by sampling $\theta$ in $[0, 2\pi]$. We vary the width of the MLP from 1024 to 8192. We adopt the Identity matrix $I$ (i.e., vanilla gradient) and a series of $S, \tilde{S}, \tilde{S}_e$ that $start = 1$ and

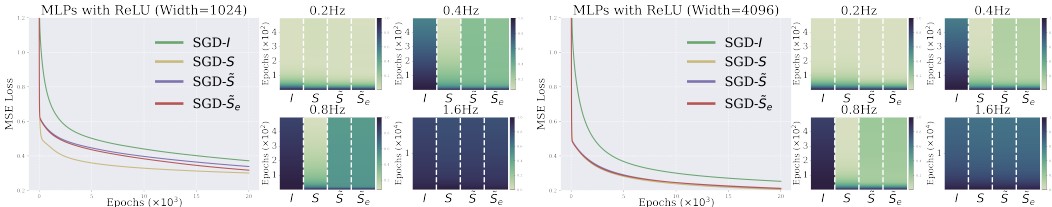

Figure 1: Evolution of approximation error with training iterations on time domain and Fourier domain. Line plots visualize the MSE loss curves of MLPs with $1024$ and $4096$ neurons optimized by four methods. Heatmaps show the relative error $\Delta_k$ on four frequency bands.

$end$ ranges from 11 to 15. The groups of our IGA method are partitioned by the order of $\theta$ and the size is 8, indicating that $\tilde{S}_e$ is only $1/64$ the size of $S, \tilde{S}$. All MLPs are trained separately with the same fixed learning rates by SGD for 20000 iterations. We visualize results of baseline model and adjustments with $end = 15$ in Fig. 1. More results can be found in our Appendix C.

**Analysis of Experiment 1.** In Fig. 1, line plots on a gray background show the convergence trends of MLPs by NTK-based ($S$), eNTK-based ($\tilde{S}$) and NTK-based ($\tilde{S}_e$) gradient adjustments. Vanilla gradient ($I$) is also compared. The heat maps with a consistent color scale visualize their impacts on the spectrum, where darker colors indicate higher errors. Note that initial colorbars of vanilla gradients ($I$) darken with increasing frequency index. This means that the MLP optimized by vanilla gradient suffers severe spectral bias. With gradient adjustments, there are more lighter shades over the second and third largest frequency, indicating that these adjustments effectively improve the spectral bias. Therefore, MLPs with gradient adjustments have faster convergence rate in MSE loss as shown in line plots. Please note that although adjustments by $S$, $\tilde{S}$ and $\tilde{S}_e$ exhibit slight difference in trends and impacts on spectrum when the network width is 1024, the difference is barely noticeable when the width is 4096. This is consistent with our Theorem 4.1 and the analysis of our Theorem 4.2. Moreover, despite the presence of difference when the network width is 1024, $\tilde{S}$ and $\tilde{S}_e$ effectively improve spectral bias and the approximation accuracy, indicating that $\tilde{K}$ links spectral bias with training dynamics and our IGA method is effective.

**Experiment 2.** In this experiment, we aims to show that our IGA method can tailor impacts on spectral bias of general architectures by managing the number of balanced eigenvalues and analyze tailored impacts under varying group sizes. Therefore, we apply our method on two practical INR models, i.e., MLPs with ReLU (ReLU) and MLPs with Sin (SIREN) (Sitzmann et al., 2020) like Shi et al. (2024a). To analyze impacts under varying sizes, we draw on settings in Rahaman et al. (2019) and construct a 1D function $f : \mathbb{R}^1 \to \mathbb{R}^1$ with abundant spectrum as follows:

$$f(x) = sin(20\pi x) + sin(40\pi x) + sin(60\pi x) + sin(80\pi x) + sin(100\pi x) + sin(120\pi x). \quad (9)$$

SIREN is trained to regress the $f(x)$ with 2048 discrete values uniformly sampled in the interval $[-1, 1]$. For ReLU, we halve the frequency of each component due to its limited representation capacity (Yüce et al., 2022). For our IGA method, we partition data points using two interval lengths of 4 and 8 to compute $\tilde{K}_e$ and varies $end$ from 2 to 8 to construct a series of $\tilde{S}_e$. ENTK-based gradient adjustment is considered. All gradient adjustments are conducted across multiple MLPs with a four-hidden-layer, 256-width architecture, optimized by Adam for 10,000 iterations with a fixed learning rate of $5e - 5$.

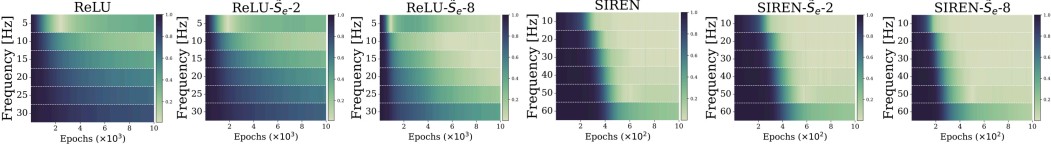

Figure 2: Progressively amplified impacts on spectral bias of ReLU and SIREN by increasing the number of balanced eigenvalues of $\tilde{S}_e$ when the group size is 8. ReLU denotes that the MLP with ReLU optimized using vanilla gradients; ReLU-$\tilde{S}_e$-2 denotes that the MLP with ReLU optimized using gradients adjusted by $\tilde{S}_e$ with $end = 2$. More results can be found in our Appendix C.

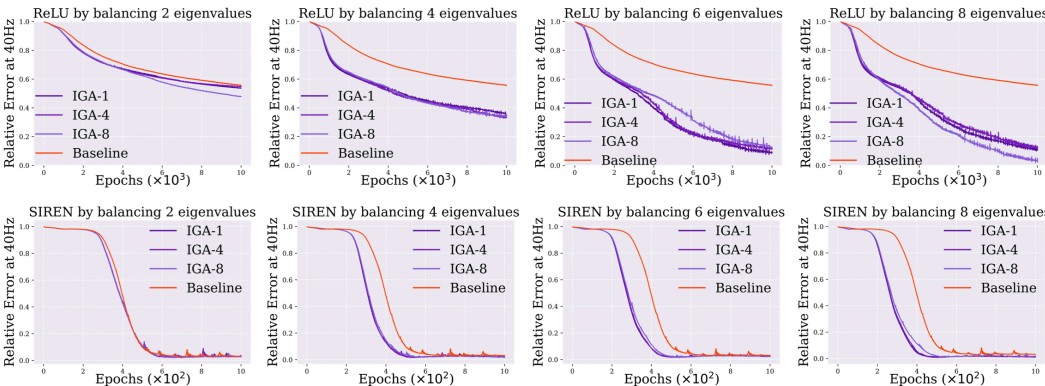

Figure 3: Comparison of group sizes with varying balanced eigenvalues on Relative Error at 40 Hz. IGA-1 denotes that the group size is 1, i.e., the eNTK-based gradient adjustment; IGA-4 denotes that the group size is 4; Baseline denotes that the MLP is optimized by vanilla gradients. Results on other frequencies can be found in Appendix C.

**Analysis of Experiment 2.** In Fig. 2 and 3, with more balanced eigenvalues, the proportion of lighter regions in heatmaps of both ReLU and SIREN grows, while the curve of relative error decreases more rapidly. These observations clearly illustrate that impacts on spectral bias by our IGA method can be amplified by increasing the number of balanced eigenvalues. Further, as shown in Fig. 3, despite the relative error curves of ReLU exhibit subtle differences as the group size increases, our IGA method effectively amplifies impacts on spectral bias by increasing the number of balanced eigenvalues.

## 6 EXPERIMENT ON VISION APPLICATIONS

In this section, we apply our IGA method to various practical applications of INRs in computer vision, demonstrating the superiority of IGA.

### 6.1 2D COLOR IMAGE APPROXIMATION

It is well known that natural images simultaneously encompass rich low- and high-frequency components (Chan & Shen, 2005). Therefore, single natural image fitting has become an ideal test bed for INR models (Sitzmann et al., 2020; Saragadam et al., 2023; Xie et al., 2023; Shi et al., 2024a). Meanwhile, spatial correlations and local features repetition of natural images are also embedded in the complexity of this image, indicating that the use of inductive generalization is reasonable.

In this experiment, we attempt to parameterize the function $\phi : \mathbb{R}^2 \to \mathbb{R}^3, \mathbf{x} \to \phi(\mathbf{x})$ that represents a discrete image. Following the previous work (Saragadam et al., 2023), we establish four-layer MLPs with 256 hidden features. We conduct experiments on three MLPs archetectures, i.e., MLPs with ReLU activation function (ReLU), ReLU with Positional encoding (PE) (Tancik et al., 2020) and MLPs with periodic activation function Sine (SIREN) (Sitzmann et al., 2020). These three models are classic baselines in the field of INRs and are widely used for comparison (Sitzmann et al., 2020; Saragadam et al., 2023; Shi et al., 2024a). We test on the first 8 images from the Kodak 24 dataset, each with a resolution of $768 \times 512$ pixels. For these images, $\boldsymbol{K}$ has approximately $10^{11}$ entries, rendering decomposition and multiplication infeasible during training. For our IGA method, we partition each image into non-overlapping $32 \times 32$ patches as groups and points with the largest residuals are sampled to form the sample set $\boldsymbol{X}_e$. We construct the corresponding transformation matrix $\tilde{\boldsymbol{S}}_e$ with $end = 20$ for SIREN and PE and $end = 25$ for ReLU due to its severe spectral bias. Therefore, we generalize the inductive gradient adjustments of less than $0.1\%$ of total data points. For all architectures, we train baseline with vanilla gradients and with our inductive gradient adjustments (+IGA) by Adam optimizer. The learning rate schedule follows (Shi et al., 2024a), which maintains a fixed rate for the first 3000 iterations and then reduces it by 0.1 for another 7000 iterations. We set initial learning rates as $5e-3$ for ReLU activation function and $1e-3$ for Sine activation function. For all baseline models, we set initial learning rates as $1e-3$ due to poor performances observed with $5e-3$. Current training adjustments methods, i.e., Fourier reparameterization (+FR) (Shi et al., 2024a) and batch normalization (+BN) (Cai et al., 2024) are

| PE (24.60dB) | PE+FR (26.63dB) | PE+BN (27.42dB) | PE+IGA (28.06dB) |

Figure 4: Visual examples of 2D color image approximation results by different training dynamics methods. Enlarged views of the regions labeled by red boxes are provided. The residuals of these regions in the Fourier domain are visualized through heatmaps at the top right corner. The increase in error corresponds to the transition of colors in the heatmaps from blue to red.

also compared. Their hyperparameters follow publicly available codes and we make every effort to achieve optimal performance. Full-batch training is adopted. In Table 1 , we report three average metrics of different INRs over the first 8 images of Kodak 24, where LPIPS values are measured by the 'alex' from Zhang et al. (2018).

As shown in Fig. 4, PE with our IGA method not only achieves best improvements on PSNR values comparing to PE, PE+FR and PE+BN, but also facilitates a more precise representation of high-frequency details instead of being overly smooth like other methods. Concretely, as shown in the spectra at the top right corner of Fig. 4, improvements of IGA are uniformly distributed across most frequency bands. In contrast, while BN and FR increase PSNR values, most improvements are near the origin, i.e., in the low-frequency range. This observation indicates that our IGA method allows for more balanced convergence rates of a wide spectral range with millions of data points. More visualization can be found in our Appendix D.

Table 1: Average metrics of 2D color image approximation results by different methods. The detailed settings can be found in Sec. 6.1. Per-image results are provided in our Appendix D.

| Average Metric | ReLU | | | | PE | | | | SIREN | | | |
|---|---|---|---|---|---|---|---|---|---|---|---|---|
| | Vanilla | +FR | +BN | +IGA | Vanilla | +FR | +BN | +IGA | Vanilla | +FR | +BN | +IGA |
| PSNR ↑ | 21.78 | 22.14 | 22.55 | **23.00** | 28.64 | 29.74 | 31.65 | **32.46** | 32.65 | 32.61 | 32.35 | **33.48** |
| SSIM ↑ | 0.483 | 0.492 | 0.505 | **0.513** | 0.783 | 0.817 | 0.870 | **0.882** | 0.898 | 0.899 | 0.894 | **0.912** |
| LPIPS$^{alex}$↓ | 0.630 | 0.631 | 0.595 | **0.555** | 0.222 | 0.187 | 0.114 | **0.090** | 0.081 | 0.081 | 0.102 | **0.067** |

## 6.2 3D SHAPE REPRESENTATION

3D shape representation by Signed Distanced Functions (SDFs) has the advantage of modeling complex topologies. Therefore, representing SDFs has been widely adopted to test INRs models (Sitzmann et al., 2020; Saragadam et al., 2023; Cai et al., 2024; Shi et al., 2024a). In this section, we evaluate IGA on this task. Five 3D objects from the public dataset (Martel et al., 2021; Zhu et al., 2024) are utilized by previous settings of (Saragadam et al., 2023; Shi et al., 2024a), which sample data points over a $512^3$ grid. We unfold these data points dimensionally and partition them into numerous groups, each containing 256 points. For each iteration, we randomly draw 512 groups and generalize the inductive gradient adjustments on these groups. The sampling strategy and MLP architecture are consistent with Experiment 6.1. We use Adam optimizer to minimize the $\ell_2$ loss between voxel values and INRs approximations. For fair comparison, the same training strategy are adopted for FR, BN and IGA. The detailed training strategy can be found in our Appendix E.

Table 2: Intersection over Union (IOU) of 3D shape representation by different methods. The detailed settings can be found in Sec. 6.1. The results for each scenario are provided in the Appendix.

| Average Metric | ReLU | | | | PE | | | | SIREN | | | |
|---|---|---|---|---|---|---|---|---|---|---|---|---|
| | Vanilla | +FR | +BN | +IGA | Vanilla | +FR | +BN | +IGA | Vanilla | +FR | +BN | +IGA |
| IOU ↑ | 9.647e-1 | 9.654e-1 | 9.542e-1 | **9.733e-1** | 9.942e-1 | 9.961e-2 | 9.938e-1 | **9.970e-1** | 9.889e-1 | 9.866e-1 | 9.825e-1 | **9.897e-1** |

In Table 2, we report the average intersection over union (IOU) metrics of five objects for reference. Under our training settings, baseline models optimized by vanilla gradient has converged to a favorable optimum, significantly outperforming prior works (Saragadam et al., 2023; Cai et al., 2024; Shi et al., 2024a). Nevertheless, our IGA method enables models to explore superior optima by purposefully improving the spectral bias, thereby leading to further improvements in representation accuracy such as more sharper edges. More results in our Appendix E.

Figure 5: Visual examples of 3D shape representation results by different training dynamics methods. Five images on the right correspond to the enlarged views of the red-boxed area of five models.

## 6.3 LEARNING 5D NEURAL RADIANCE FIELDS

Learning neural radiance fields for novel view synthesis, i.e., NeRF, is the main application of INRs (Mildenhall et al., 2021; Saragadam et al., 2023; Xie et al., 2023; Shi et al., 2024a). The main process of NeRF is to build a neural radiance field from a 5D coordinate space to RGB space by a MLP $f(\mathbf{x}, \Theta)$. Specifically, given a ray $i$ from the camera into the scene, the MLP takes the 5D coordinates (3D spatial locations and 2D view directions) of $N$ points along the ray as the inputs and outputs the corresponding color $\mathbf{c}$ and volume density $\sigma$. Then $\mathbf{c}$ and $\sigma$ are combined using numerical volume rendering to obtain the final pixel color of the ray $i$. We set each ray as one group . For each group, we sample the point with the maximum integral weight. We apply our method to the original NeRF (Mildenhall et al., 2021). The "NeRF-pytorch" codebase (Yen-Chen, 2020) is used and we follow its default settings for all methods. More details can be found in Appendix F.

Table 3 lists average metrics of four methods on the down-scaled Blender dataset (Mildenhall et al., 2021). Our methods achieves the best results among these methods. The average improvement of our method is up to 0.21dB, which is nearly twice that of FR and BN. Visualization results in our Appendix F show that our IGA enables NeRF to capture more accurate and high-frequency reconstruction results, thereby improving PSNR values.

Table 3: Average metrics of 5D neural radiance fields by different methods. Detailed settings can be found in Sec. 6.3. Per-scene results are provided in our Appendix F.

| Metrics | NeRF | NeRF+FR | NeRF+BN† | NeRF+TDS |
|---|---|---|---|---|
| PSNR ↑ | 31.23 | 31.35 | 31.37 | 31.47 |
| SSIM ↑ | 0.953 | 0.954 | 0.956 | 0.955 |
| LPIPS ↓ | $0.029^{\text{alex}}$ | $0.028^{\text{alex}}$ | 0.050 | $0.027^{\text{alex}}$ |

† Use values reported on the paper (Cai et al., 2024).

## 7 CONCLUSION

In this paper, we propose an effective gradient adjustment strategy to purposefully improve the spectral bias of multi-layer perceptrons (MLPs) for better implicit neural representations (INRs). We delve into the linear dynamics model of MLPs and theoretically identify that the empirical Neural Tangent Kernel (eNTK) matrix connect spectral bias with the linear dynamics of MLPs. Based on eNTK matrix, we propose our inductive gradient adjustment method via inductive generalization of gradient adjustments from sampled data points. Both theoretical and empirical analysis are conducted to validate impacts of our method on spectral bias. Further, we validate our method on various real-world vision applications of INRs. Our method can effectively tailor improvements on spectral bias and lead to better representation for common INRs network architectures. We hope our study could inspire future works to focus on controlling the bias of neural networks by adjusting training dynamics and yield better performance.

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

# A   PROOFS FOR SECTION 4

In this section, we give proofs of theorem in the section 4 and more detailed explanation of our theoretical results.

## A.1   ANALYSIS FRAMEWORK

For the sake of simplicity and to focus on the core aspects of the problem, we follow the previous works (Lee et al., 2019; Geifman et al., 2023; Arora et al., 2019), and perform a theoretical analysis on the basis of the following framework: assuming that the training set $\{\boldsymbol{x}_i, y_i\}_{i=1}^{N}$ is contained in some compact set, a two-layer network $f(\boldsymbol{x}; \Theta)$ with $m$ neurons is formalized as follows to fit these data:

$$f(\boldsymbol{x}; \Theta) = \frac{1}{\sqrt{m}} \sum_{r=1}^{m} a_r \sigma(\boldsymbol{w}_r^\top \boldsymbol{x} + b_r), \tag{10}$$

where the activation function $\sigma$ satisfies that $|\sigma(0)|, \|\sigma'\|_\infty, sup_{x \neq x'}|\sigma'(x) - \sigma'(x')|/|x - x'| < \infty$. The parameters of $f(\boldsymbol{x}; \Theta)$ are randomly initialized with $\mathcal{N}(0, \frac{c_\sigma}{m})$, except for biases initialized with $\mathcal{N}(0, c_\sigma)$, where $c_\sigma = 1/\mathbb{E}_{z \sim \mathcal{N}(0,1)}[\sigma(z)^2]$. The loss function is $\frac{1}{2} \sum_{i=1}^{N}(f(\boldsymbol{x}_i, \Theta_t) - y_i)^2$. Then, the NTK matrix $\boldsymbol{K}$ and the empirical NTK matrix $\tilde{\boldsymbol{K}}$ is defined as follows:

$$\boldsymbol{K} = \mathbb{E}_{\Theta_0}[\nabla_{\Theta_0} f(\boldsymbol{X}; \Theta_0)^\top \nabla_{\Theta_0} f(\boldsymbol{X}; \Theta_0)] \tag{11}$$

$$\tilde{\boldsymbol{K}} = \nabla_{\Theta_t} f(\boldsymbol{X}; \Theta_t)^\top \nabla_{\Theta_t} f(\boldsymbol{X}; \Theta_t), \tag{12}$$

where $\nabla_\Theta f(\boldsymbol{X}; \Theta) \in \mathbb{R}^{q \times N}$; $q$ is the number of parameters. We assume that $\boldsymbol{K}, \tilde{\boldsymbol{K}}$ are full rank indicating that they are positive definite. Then we have the following standard orthogonal spectral decomposition: $\boldsymbol{K} = \sum_{i=1}^{N} \lambda_i \boldsymbol{v}_i \boldsymbol{v}_i^\top$ and $\tilde{\boldsymbol{K}} = \sum_{i=1}^{N} \tilde{\lambda}_i \tilde{\boldsymbol{v}}_i \tilde{\boldsymbol{v}}_i^\top$, which eigenvalues are indexed in descending order of magnitude. We construct the corresponding transformation matrices as $\boldsymbol{S} = \sum_{i=1}^{N} \frac{g(\lambda_i)}{\lambda_i} \boldsymbol{v}_i \boldsymbol{v}_i^\top$ and $\tilde{\boldsymbol{S}} = \sum_{i=1}^{N} \frac{g(\tilde{\lambda}_i)}{\tilde{\lambda}_i} \tilde{\boldsymbol{v}}_i \tilde{\boldsymbol{v}}_i^\top$. From Jacot et al. (2018); Du et al. (2018); Arora et al. (2019); Geifman et al. (2023), we have that:

$$\lim_{m \to \infty} \|\boldsymbol{K} - \tilde{\boldsymbol{K}}\|_F = 0. \tag{13}$$

## A.2   BASIC LEMMAS

In this section, we show some basic lemmas. Lemma 1 describes the local properties of $f(\boldsymbol{x}, \Theta)$. Based on Lemma 1, we derive Lemma 2, 3 and 4, which characterize training dynamics of $f(\boldsymbol{x}, \Theta)$ under different adjusted gradients.

**Lemma 1.** *(Modified from Lemma A.1 of Geifman et al. (2023)) For bounded matrices, i.e., $\boldsymbol{I}, \boldsymbol{S}, \tilde{\boldsymbol{S}}$, there is a $\kappa > 0$ such that for every $C > 0$, with high probability over random initialization the following holds: $\forall \Theta, \tilde{\Theta} \in B(\Theta_0, Cm^{-\frac{1}{2}})$ at time step $t$:*

$$\|(\nabla_\Theta f(\boldsymbol{X}) - \nabla_{\tilde{\Theta}} f(\boldsymbol{X}))\boldsymbol{A}\|_F \leq \frac{\kappa}{\sqrt{m}} \|\Theta - \tilde{\Theta}\|_F \tag{14}$$

$$\|\nabla_\Theta f(\boldsymbol{X})\boldsymbol{A}\|_F \leq \frac{\kappa}{\sqrt{m}} \tag{15}$$

*where $\boldsymbol{A}$ can be $\boldsymbol{I}, \boldsymbol{S}, \tilde{\boldsymbol{S}}$.*

As discussed in Lee et al. (2019); Geifman et al. (2023), the core of Lemma 1 is the requirement that matrix $A$ is bounded. Therefore, Lemma 1 clearly holds.

**Lemma 2.** *The parameters are updated by: $\Theta_{t+1} = \Theta_t - \eta \nabla_{\Theta_t} f(\boldsymbol{X}, \Theta_t)\boldsymbol{S}\boldsymbol{r}_t$ with $\eta < \frac{2}{min(g(\lambda))+max(g(\lambda))}$. For $\epsilon > 0$, there always exists $M > 0$, when $m > M$, such that with high probability over the random initialization, we have that :*

$$\|\boldsymbol{r}_{t+1}\|_2^2 = \sum_{i=1}^{N}(1 - \eta g(\lambda_i))^2(\boldsymbol{v}_i^\top \boldsymbol{r}_t)^2 \pm \xi(t), \tag{16}$$

*where $\boldsymbol{r}_t = (f(\boldsymbol{x}_i, \Theta_t) - y_i)_{i=1}^{N}$ and $|\xi(t)| < \epsilon$*

*Proof.* By the mean value theorem with respect to parameters $\Theta$, we can have that:

$$\boldsymbol{r}_{t+1} = \boldsymbol{r}_{t+1} - \boldsymbol{r}_t + \boldsymbol{r}_t = \nabla_{\Theta_t'} f(\boldsymbol{X})^\top (\Theta_{t+1} - \Theta_t) + \boldsymbol{r}_t = \nabla_{\Theta_t'} f(\boldsymbol{X})^\top (-\eta \nabla_{\Theta_t} f(\boldsymbol{X}) \boldsymbol{S} \boldsymbol{r}_t) + \boldsymbol{r}_t$$

$$= (\boldsymbol{I} - \eta \boldsymbol{K} \boldsymbol{S}) \boldsymbol{r}_t + \underbrace{\eta (\boldsymbol{K} - \tilde{\boldsymbol{K}}) \boldsymbol{S} \boldsymbol{r}_t}_{A} + \underbrace{\eta (\nabla_{\Theta_t} f(\boldsymbol{X}) - \nabla_{\Theta_t'} f(\boldsymbol{X}))^\top \nabla_{\Theta_t} f(\boldsymbol{X}) \boldsymbol{S} \boldsymbol{r}_t}_{B},$$

where $\Theta_t'$ lies on the line segment $\overline{\Theta_t \Theta_{t+1}}$. We define that $\boldsymbol{\xi}'(t) = A + B$. For $\|A\|$, we have that:

$$\|A\| \leq \eta \|(\boldsymbol{K} - \tilde{\boldsymbol{K}}) \boldsymbol{S} \boldsymbol{r}_t\| \leq \eta \|\boldsymbol{K} - \tilde{\boldsymbol{K}}\|_F \|\boldsymbol{S}\|_F \|\boldsymbol{r}_t\|_2$$

$$\leq^{(1)} \eta L R_0 \|\boldsymbol{K} - \tilde{\boldsymbol{K}}\|_F \leq^{(2)} \frac{\epsilon}{2}$$

where (1) follows [Geifman et al. (2023)](#) that $\boldsymbol{S}$ and $\boldsymbol{r}_t$ are bounded by constants $L$ and $R_0$, respectively; (2) follows the equation [13](#).

For $\|B\|$, we have that:

$$\|B\| \leq \eta \|\nabla_{\Theta_t} f(\boldsymbol{X}, \Theta_t) - \nabla_{\Theta_t'} f(\boldsymbol{X}, \Theta_t')\|_F \|\nabla_{\Theta_t} f(\boldsymbol{X}) \boldsymbol{S}\|_F \|\boldsymbol{r}_t\|_2 \leq \frac{\eta \kappa^2 R_0}{m} \|\Theta_t - \Theta_t'\|_2$$

$$\leq \frac{\eta \kappa^2 R_0}{m} \|\Theta_t - \Theta_{t+1}\|_2 \leq \frac{\eta \kappa^2 R_0}{m} \|\eta \nabla_{\Theta_t} f(\boldsymbol{X}, \Theta_t) \boldsymbol{S} \boldsymbol{r}_t\| \leq^{(1)} \frac{\eta^2 \kappa^3 R_0^2}{m^{3/2}}$$

where (1) follows the Lemma [1](#). Therefore, we have that $\boldsymbol{r}_{t+1} = \sum_{i=1}^N (1 - \eta g(\lambda_i)) (\boldsymbol{v}_i^\top \boldsymbol{r}_t) \boldsymbol{v}_i \pm \boldsymbol{\xi}'(t)$; there always exits $M_1 > 0$, such that $m > M_1$, $\|\boldsymbol{\xi}'(t)\| \leq \|A\| + \|B\| < \epsilon$. Further, we have that:

$$\|\boldsymbol{r}_{t+1}\|_2^2 = \sum_{i=1}^N (1 - \eta g(\lambda_i))^2 (\boldsymbol{v}_i^\top \boldsymbol{r}_t)^2 + \|\boldsymbol{\xi}'(t)\|_2^2 \pm 2\boldsymbol{\xi}'(t)^\top \sum_{i=1}^N (1 - \eta g(\lambda_i)) (\boldsymbol{v}_i^\top \boldsymbol{r}_t) \boldsymbol{v}_i$$

$$= \sum_{i=1}^N (1 - \eta g(\lambda_i))^2 (\boldsymbol{v}_i^\top \boldsymbol{r}_t)^2 + \underbrace{\|\boldsymbol{\xi}'(t)\|_2^2 \pm 2\boldsymbol{\xi}'(t)^\top (\boldsymbol{r}_t \pm \boldsymbol{\xi}'(t))}_{\xi(t)}$$

For $\xi(t)$, as $\boldsymbol{r}_{t+1}$ is bounded by $R_0$, there always exists $M > M_1$, such that:

$$|\xi(t)| \leq 3\|\boldsymbol{\xi}'(t)\|_2^2 + 2R_0 \|\boldsymbol{\xi}'(t)\|_2^2 < \epsilon.$$

$\square$

**Lemma 3.** *The parameters are updated by:* $\Theta_{t+1} = \Theta_t - \eta \nabla_{\Theta_t} f(\boldsymbol{X}, \Theta_t) \tilde{\boldsymbol{S}} \boldsymbol{r}_t$ *with* $\eta = \frac{2}{max(g(\tilde{\lambda})) + min(g(\tilde{\lambda}))}$. *For* $\epsilon > 0$, *there always exists* $M > 0$, *when* $m > M$, *such that with high probability over the random initialization , we have that :*

$$\|\tilde{\boldsymbol{r}}_{t+1}\|_2^2 = \sum_{i=1}^N (1 - \eta g(\tilde{\lambda}_i))^2 (\tilde{\boldsymbol{v}}_i^\top \boldsymbol{r}_t)^2 \pm \tilde{\xi}(t), \tag{17}$$

*where* $\boldsymbol{r}_t = (f(\boldsymbol{x}_i, \Theta_t) - y_i)_{i=1}^N$ *and* $|\tilde{\xi}(t)| < \epsilon$

*Proof.* By the mean value theorem with respect to parameters $\Theta$, we can have that:

$$\boldsymbol{r}_{t+1} = \boldsymbol{r}_{t+1} - \boldsymbol{r}_t + \boldsymbol{r}_t = \nabla_{\Theta_t'} f(\boldsymbol{X}, \Theta_t')^\top (\Theta_{t+1} - \Theta_t) + \boldsymbol{r}_t = \nabla_{\Theta_t'} f(\boldsymbol{X}, \Theta_t')^\top (-\eta \nabla_{\Theta_t} f(\boldsymbol{X}, \Theta_t) \tilde{\boldsymbol{S}} \boldsymbol{r}_t) + \boldsymbol{r}_t$$

$$= (\boldsymbol{I} - \eta \boldsymbol{K}^t \tilde{\boldsymbol{S}}) \boldsymbol{r}_t + \underbrace{\eta (\nabla_{\Theta_t} f(\boldsymbol{X}, \Theta_t) - \nabla_{\Theta_t'} f(\boldsymbol{X}, \Theta_t'))^\top \nabla_{\Theta_t} f(\boldsymbol{X}) \tilde{\boldsymbol{S}} \boldsymbol{r}_t}_{\tilde{\boldsymbol{\xi}}'(t)}.$$

For $\tilde{\boldsymbol{\xi}}'(t)$, we follow the same technique in the proof of [2](#). Then we can have that:

$$\|\tilde{\boldsymbol{\xi}}'(t)\| \leq \eta \|\nabla_{\Theta_t} f(\boldsymbol{X}) - \nabla_{\Theta_t'} f(\boldsymbol{X})\|_F \|\nabla_{\Theta_t} f(\boldsymbol{X}) \boldsymbol{S}^t\|_F \|\boldsymbol{r}_t\|_2 \leq \eta \frac{\kappa^2}{m} \|\Theta_t - \Theta_t'\|_2$$

$$\leq \frac{\eta \kappa^2 R_0}{m} \|\Theta_t - \Theta_{t+1}\|_2 \leq \frac{\eta^2 \kappa^2 R_0^2}{m^{3/2}}$$

Therefore, we have that $\boldsymbol{r}_{t+1} = \sum_{i=1}^{N}(1-\eta g(\tilde{\lambda}_i))(\tilde{\boldsymbol{v}}_i^\top \boldsymbol{r}_t)\tilde{\boldsymbol{v}}_i \pm \tilde{\boldsymbol{\xi}}'(t)$; there always exits $M_1 > 0$, such that $m > M_2$, $\|\tilde{\boldsymbol{\xi}}'(t)\| < \epsilon$. Further, we have that:

$$\|\boldsymbol{r}_{t+1}\|_2^2 = \sum_{i=1}^{N}(1-\eta g(\tilde{\lambda}_i))^2(\tilde{\boldsymbol{v}}_i^\top \boldsymbol{r}_t)^2 + \|\tilde{\boldsymbol{\xi}}'(t)\|_2^2 \pm 2\tilde{\boldsymbol{\xi}}'(t)^\top \sum_{i=1}^{N}(1-\eta g(\tilde{\lambda}_i))(\tilde{\boldsymbol{v}}_i^\top \boldsymbol{r}_t)\tilde{\boldsymbol{v}}_i$$

$$= \sum_{i=1}^{N}(1-\eta g(\tilde{\lambda}_i))^2(\tilde{\boldsymbol{v}}_i^\top \boldsymbol{r}_t)^2 + \underbrace{\|\tilde{\boldsymbol{\xi}}'(t)\|_2^2 \pm 2\tilde{\boldsymbol{\xi}}'(t)^\top(\boldsymbol{r}_t \pm \tilde{\boldsymbol{\xi}}'(t))}_{\tilde{\xi}(t)}$$

For $\xi(t)$, as $\boldsymbol{r}_{t+1}$ is bounded by $R_0$, there always exists $M > M_2$, such that:

$$|\tilde{\xi}(t)| \le 3\|\tilde{\boldsymbol{\xi}}'(t)\|_2^2 + 2R_0\|\tilde{\boldsymbol{\xi}}'(t)\|_2^2 < \epsilon.$$

$\square$

**Lemma 4.** *The parameters are updated by vanilla gradient descent. For $\epsilon > 0$, there always exists $M > 0$, when $m > M$, such that with high probability over the random initialization, we have that:*

$$f(\boldsymbol{x}_i, \Theta_{t+1}) - f(\boldsymbol{x}_i, \Theta_t) = -\eta\nabla_{\Theta_t}f(\boldsymbol{x}_i, \Theta_t)^\top \sum_{j=1}^{N}\nabla_{\Theta_t}f(\boldsymbol{x}_j, \Theta_t)(f(\boldsymbol{x}_j, \Theta_t) - y_j) \pm \xi_i'(t), \quad (18)$$

*where $|\xi_i'(t)| < \epsilon$.*

*Proof.* By the mean value theorem with respect to parameters $\Theta$, we can have that:

$$f(\boldsymbol{x}_i, \Theta_{t+1}) = \nabla_{\Theta_t'}f(\boldsymbol{x}_i, \Theta_t')^\top(\Theta_{t+1} - \Theta_t) + f(\boldsymbol{x}_i, \Theta_t)$$

$$= \nabla_{\Theta_t'}f(\boldsymbol{x}_i, \Theta_t')^\top[-\eta\sum_{j=1}^{N}\nabla_{\Theta_t}f(\boldsymbol{x}_i, \Theta_t)(f(\boldsymbol{x}_j, \Theta_t) - y_j)] + f(\boldsymbol{x}_i, \Theta_t)$$

$$= -\eta\nabla_{\Theta_t}f(\boldsymbol{x}_i, \Theta_t)^\top \sum_{j=1}^{N}\nabla_{\Theta_t}f(\boldsymbol{x}_j, \Theta_t)(f(\boldsymbol{x}_j, \Theta_t) - y_j) + f(\boldsymbol{x}_i, \Theta_t) + \xi_i'(t).$$

For $\xi_i'(t)$, we have that:

$$|\xi_i'(t)| = |(\nabla_{\Theta_t'}f(\boldsymbol{x}_i, \Theta_t') - \nabla_{\Theta_t}f(\boldsymbol{x}_i, \Theta_t))^\top \sum_{j=1}^{N}\nabla_{\Theta_t}f(\boldsymbol{x}_j, \Theta_t)(f(\boldsymbol{x}_j, \Theta_t) - y_j)|$$

$$\le \|\nabla_{\Theta_t'}f(\boldsymbol{x}_i, \Theta_t') - \nabla_{\Theta_t}f(\boldsymbol{x}_i, \Theta_t)\|_2\|\sum_{j=1}^{N}\nabla_{\Theta_t}f(\boldsymbol{x}_j, \Theta_t)(f(\boldsymbol{x}_j, \Theta_t) - y_j)\|_2$$

$$\le \|\nabla_{\Theta_t'}f(\boldsymbol{x}_i, \Theta_t') - \nabla_{\Theta_t}f(\boldsymbol{x}_i, \Theta_t)\|_2\|\nabla_{\Theta_t}f(\boldsymbol{X}, \Theta_t)\|_2\|\boldsymbol{r}_t\|_2$$

$$\le^{(1)} \frac{\kappa^2 R_0}{m}\|\Theta_{t+1} - \Theta_t\|_2 \le^{(2)} \frac{\eta\kappa^3 R_0^2}{m^{3/2}},$$

where $(1), (2)$ follow the Lemma 1 and $\boldsymbol{r}_t$ is bounded by $R_0$; Therefore, there exists $M > 0$, such that $m > M$, we have that $|\xi_i'(t)| < \epsilon$ $\square$

Lemma 2 and lemma 3 show the training dynamics of the residual $\boldsymbol{r}_t$ by $\boldsymbol{K}$-based and $\tilde{\boldsymbol{K}}$-based gradient adjustments. Equation 16 shows that $g(\lambda_i)$ controls the decay rates of different frequency components of the residual $\boldsymbol{r}_t$ as eigenvectors of $\boldsymbol{K}$ are the spherical harmonics (Ronen et al., 2019). Despite $\tilde{\boldsymbol{K}}$-based and $\boldsymbol{K}$-based gradient adjustment have the similar form, they differ in the eigenvector directions $\tilde{\boldsymbol{v}}_i$, which leads to errors. We give a theoretical analysis to this error in 4.1.

### A.3 PROOF OF THEOREM 4.1

**Theorem A.1.** *(Theorem 4.1 from the paper) The following standard orthogonal spectral decomposition exists:* $\boldsymbol{K} = \sum_{i=1}^{N} \lambda_i \boldsymbol{v}_i \boldsymbol{v}_i^\top$ *and* $\tilde{\boldsymbol{K}} = \sum_{i=1}^{N} \tilde{\lambda}_i \tilde{\boldsymbol{v}}_i \tilde{\boldsymbol{v}}_i^\top$, *satisfying that* $\boldsymbol{v}_i^\top \tilde{\boldsymbol{v}}_i > 0$ *for* $i \in [N]$. *We denoted as* $max_i\{min\{\lambda_i - \lambda_{i+1}, \lambda_{i+1} - \lambda_{i+2}\}\} = G$. *Let* $\{g_i(x)\}_{i=1}^{N}$ *be a set of Lipschitz continuous functions, with the Supremum of their Lipschitz constants denoted by* $k$, *with* $\eta < min\{(max(g(\lambda)) + min(g(\lambda)))^{-1}, (max(g(\tilde{\lambda})) + min(g(\tilde{\lambda})))^{-1}\}$, *for* $\epsilon > 0$, *there always exists* $M > 0$, *such that* $m > M$, *for* $i \in [N]$, *we have that* $|g(\lambda_i) - g(\tilde{\lambda}_i)| < \epsilon_1, \|\boldsymbol{v}_i - \tilde{\boldsymbol{v}}_i\| < \epsilon_2$; *furthermore, we have that:*

$$|(1 - \eta g(\tilde{\lambda}_i))^2 (\tilde{\boldsymbol{v}}_i^\top \boldsymbol{r}_t)^2 - (1 - \eta g(\lambda_i))^2 (\boldsymbol{v}_i^\top \boldsymbol{r}_t)^2| < \epsilon_3, \|\boldsymbol{r}_{t+1} - \tilde{\boldsymbol{r}}_{t+1}\| < \epsilon_4,$$

*where* $\epsilon_1 = k\epsilon$; $\epsilon_2 = (2^{3/2}\epsilon)/G$; $\epsilon_3 = \frac{8R_0}{G^2}(k\eta\epsilon^3 + (v+1)\epsilon^2 + (v^2 + Gv)\epsilon)$; $\epsilon_4 = \epsilon N((\frac{16vR_0}{G} + \eta kv^2 R_0) + \epsilon^2) + 2\epsilon$; $v, k, R_0$ *are constants.*

*Proof.* We define that the perturbation of $\boldsymbol{K}$ as $\boldsymbol{E} = \boldsymbol{K} - \tilde{\boldsymbol{K}}$. As previously discussed, we have that $\lim_{m \to \infty} \|\boldsymbol{K} - \tilde{\boldsymbol{K}}\|_F = 0$. Therefore, for any $0 < \epsilon$ there always exists $M_0$, such that $m > M_0$, we have that $\|\boldsymbol{E}\|_F < \epsilon$. As $\boldsymbol{K}, \tilde{\boldsymbol{K}}$ are real symmetric matrices, $\boldsymbol{E}$ is still a real symmetric matrix. This indicates that:

$$\|\boldsymbol{E}\|_2 = \sqrt{\lambda_1'^2} \leq \sqrt{\sum_{i=1}^{N} \lambda_i'^2} = \sqrt{trace(\boldsymbol{E}^\top \boldsymbol{E})} = \|\boldsymbol{E}\|_F$$

where $\lambda_i'$ is the largest eigenvalue of $\boldsymbol{E}$.

By Wely's inequality, when $m > M_0$, for $i = 1, \ldots, N$, we have that $|\lambda_i - \tilde{\lambda}_i| \leq \|\boldsymbol{E}\|_2 < \epsilon$. As $g_i(\lambda)$ is a Lipschitz continuous function with the Lipschitz constant $k$, for $i \in [N]$, we have that:

$$|g(\lambda_i) - g(\tilde{\lambda}_i)| \leq k|\lambda_i - \tilde{\lambda}_i| < k\epsilon$$

Therefore, for any $0 < \epsilon$, there always exists $M_0$, such that $|g(\lambda_i) - g(\tilde{\lambda}_i)| < k\epsilon$. By the Corollary 3. of Yu et al. (2015) that a variant of Davis-Kahan theorem, we have that:

$$\|\tilde{\boldsymbol{v}}_i - \boldsymbol{v}_i\| \leq \frac{2^{3/2}\|\boldsymbol{E}\|_2}{G} < \frac{2^{3/2}\epsilon}{G}, i \in [N]$$

Then, we have that:

$$|(1 - \eta g_i(\lambda_i))^2 (\boldsymbol{v}_i^\top \boldsymbol{r}_t)^2 - (1 - \eta g_i(\tilde{\lambda}_i))^2 (\tilde{\boldsymbol{v}}_i^\top \boldsymbol{r}_t)^2|$$

$$= |(1 - \eta g_i(\lambda_i))^2 (\boldsymbol{v}_i^\top \boldsymbol{r}_t)^2 - (1 - \eta g_i(\lambda_i))^2 (\tilde{\boldsymbol{v}}_i^\top \boldsymbol{r}_t)^2 + (1 - \eta g_i(\lambda_i))^2 (\tilde{\boldsymbol{v}}_i^\top \boldsymbol{r}_t)^2 - (1 - \eta g_i(\tilde{\lambda}_i))^2 (\tilde{\boldsymbol{v}}_i^\top \boldsymbol{r}_t)^2|$$

$$< \underbrace{|(1 - \eta g_i(\lambda_i))(\boldsymbol{v}_i^\top \boldsymbol{r}_t + \tilde{\boldsymbol{v}}_i^\top \boldsymbol{r}_t)(\boldsymbol{v}_i^\top \boldsymbol{r}_t - \tilde{\boldsymbol{v}}_i^\top \boldsymbol{r}_t)|}_{A} + \underbrace{|\eta(g_i(\tilde{\lambda}_i) - g_i(\lambda_i))(\tilde{\boldsymbol{v}}_i^\top \boldsymbol{r}_t)^2|}_{B}.$$

For $A$, we have that:

$$A < |1 - \eta g_i(\lambda_i)| \cdot \|\boldsymbol{v}_i^\top + \tilde{\boldsymbol{v}}_i^\top\|_2 \|\boldsymbol{v}_i^\top - \tilde{\boldsymbol{v}}_i^\top\|_2 \|\boldsymbol{r}_t\|_2^2$$

$$< |1 - \eta g_i(\lambda_i)| \cdot (2\|\boldsymbol{v}_i^\top\| + \|\tilde{\boldsymbol{v}}_i^\top - \boldsymbol{v}_i\|_2)\|\tilde{\boldsymbol{v}}_i^\top - \boldsymbol{v}_i\|_2 \|\boldsymbol{r}_t\|_2^2 < \epsilon \frac{8R_0^2(G\|\boldsymbol{v}_i^\top\| + \epsilon)}{G^2}$$

As $\boldsymbol{S}$ is fixed, we have that a constant $v$ to bound $\|\boldsymbol{v}_i\|$, for $i \in [N]$. Therefore, for $\epsilon > 0$, there always exists $M$, such that $m > M_0$, we have that $A < \frac{8R_0^2(\epsilon Gv + \epsilon^2)}{G^2}$.

For B, we have that:

$$B < \eta|g_i(\tilde{\lambda}_i) - g_i(\lambda_i)|(\tilde{\boldsymbol{v}}_i^\top \boldsymbol{r}_t)^2 < \epsilon k\eta(\|\tilde{\boldsymbol{v}}_i^\top - \boldsymbol{v}_i^\top\|\|\boldsymbol{r}_t\| + \|\boldsymbol{v}_i\|\|\boldsymbol{r}_t\|)^2 < \epsilon k\eta R_0(\frac{2^{3/2}}{G}\epsilon + v)^2$$

Further, we have that:

$$|(1 - \eta g_i(\lambda_i))^2 (\boldsymbol{v}_i^\top \boldsymbol{r}_t)^2 - (1 - \eta g_i(\tilde{\lambda}_i))^2 (\tilde{\boldsymbol{v}}_i^\top \boldsymbol{r}_t)^2| < \frac{8R_0}{G^2}(k\eta\epsilon^3 + (v+1)\epsilon^2 + (v^2 + Gv)\epsilon)$$

Following these analytical approaches, it is trivial to have the following result:

$$|(1 - \eta g(\lambda_i))(\boldsymbol{v}_i^\top \boldsymbol{r}_t)\boldsymbol{v}_i - (1 - \eta g(\tilde{\lambda}_i))(\tilde{\boldsymbol{v}}_i^\top \boldsymbol{r}_t)\tilde{\boldsymbol{v}}_i| < \epsilon(\frac{16vR_0}{G} + \eta k v^2 R_0) + \epsilon^2.$$

Combing this result with the Lemma 2 and 3, there always exists $M > max\{M_0, M_1, M_2\}$, we have that:

$$\|\boldsymbol{r}_{t+1} - \tilde{\boldsymbol{r}}_{t+1}\| = \|\sum_{i=1}^{N}[(1 - \eta g(\lambda_i))(\boldsymbol{v}_i^\top \boldsymbol{r}_t)\boldsymbol{v}_i - (1 - \eta g(\tilde{\lambda}_i))(\tilde{\boldsymbol{v}}_i^\top \boldsymbol{r}_t)\tilde{\boldsymbol{v}}_i] \pm \boldsymbol{\xi}'(t) \pm \tilde{\boldsymbol{\xi}}'(t)\|$$

$$< \epsilon N((\frac{16vR_0}{G} + \eta k v^2 R_0) + \epsilon^2) + \|\boldsymbol{\xi}'(t)\| + \|\tilde{\boldsymbol{\xi}}'(t)\|$$

$$< \epsilon N((\frac{16vR_0}{G} + \eta k v^2 R_0) + \epsilon^2) + 2\epsilon$$

$$\square$$

## A.4 PROOF OF THEOREM 4.2

**Theorem A.2.** *(Theorem 4.2 from the paper)* $\boldsymbol{X} = \{\boldsymbol{x}_i\}_{i=1}^N$ *are partitioned by order into* $n$ *shared groups, where each group is* $\boldsymbol{X}_j = \{\boldsymbol{x}_1^j, \ldots, \boldsymbol{x}_p^j\}$ *and* $N = np$. *For each group, one data point is sampled denoted as* $\boldsymbol{x}^j$ *and these* $n$ *data points form* $\boldsymbol{X}_e = \{\boldsymbol{x}^j\}_{j=1}^n, n \ll N$. *There exists* $\epsilon > 0$, *for* $j = 1, \ldots, n$ *and any* $1 \le i_1, i_2 \le p$, *such that* $|(f(\boldsymbol{x}_{i_1}^j, \Theta_t) - y_{i_1}^j) - (f(\boldsymbol{x}_{i_2}^j, \Theta_t) - y_{i_2}^j)|, \|\nabla_{\Theta_t} f(\boldsymbol{x}_{i_1}^j) - \nabla_{\Theta_t} f(\boldsymbol{x}_{i_2}^j)\| < \epsilon$. *Then, for any* $\boldsymbol{x}_i$, *assumed that* $\boldsymbol{x}_i$ *belongs to* $\boldsymbol{X}_{j_i}$, *such that:*

$$|\Delta f(\boldsymbol{x}_i, \Theta_t) - \nabla_{\Theta_t} f(\boldsymbol{x}^{j_i}, \Theta_t)^\top [p \sum_{j=1}^n \nabla_{\Theta_t} f(\boldsymbol{x}^j, \Theta_t)(f(\boldsymbol{x}^j) - y^j)]| < \epsilon(\frac{(\kappa + n^{3/2})R_0}{\sqrt{m}}) + \frac{\eta \kappa^3 R_0^2}{m^{3/2}},$$

*where* $\Delta f(\boldsymbol{x}_i, \Theta_t) = (\boldsymbol{r}_{t+1} - \boldsymbol{r}_t)_i$ *denotes the dynamics of* $f(\boldsymbol{x}, \Theta)$ *at time step* $t$; $R_0, \kappa$ *are constants;* $\nabla_{\Theta_t} f(\boldsymbol{x}^{j_i}, \Theta)^\top \nabla_{\Theta_t} f(\boldsymbol{x}^j, \Theta)$ *is the entry in* $\tilde{\boldsymbol{K}}_e(i, j)$.

*Proof.* From the equation 18 of lemma 4, we consider the following error:

$$|\nabla_{\Theta_t} f(\boldsymbol{x}_i, \Theta_t)^\top \sum_{i=1}^N \nabla_{\Theta_t} f(\boldsymbol{x}_i, \Theta_t)(f(\boldsymbol{x}_i, \Theta_t) - y_i) - \nabla_{\Theta_t} f(\boldsymbol{x}^{j_i}, \Theta_t)^\top [p \sum_{j=1}^n \nabla_{\Theta_t} f(\boldsymbol{x}^j, \Theta_t)(f(\boldsymbol{x}^j) - y^j)]|.$$

Following previous analysis technique, this error can be bounded by $A + B$. For $A$, we have that:

$$A = \|(\nabla_{\Theta_t} f(\boldsymbol{x}_i, \Theta_t)^\top - \nabla_{\Theta_t} f(\boldsymbol{x}^{j_i}, \Theta_t)^\top) \sum_{j=1}^N \nabla_{\Theta_t} f(\boldsymbol{x}_j, \Theta_t)(f(\boldsymbol{x}_j, \Theta_t) - y_j)\|$$

$$\le \|\nabla_{\Theta_t} f(\boldsymbol{x}_i, \Theta_t) - \nabla_{\Theta_t} f(\boldsymbol{x}^{j_i}, \Theta_t)\| \|\sum_{j=1}^N \nabla_{\Theta_t} f(\boldsymbol{x}_j, \Theta_t)(f(\boldsymbol{x}_j, \Theta_t) - y_j)\|$$

$$\le \epsilon \|\nabla_{\Theta_t} f(\boldsymbol{X}, \Theta_t)\boldsymbol{I}\boldsymbol{r}_t\| \le \epsilon\frac{\kappa R_0}{\sqrt{m}}$$

For $B$, we have that:

$$B = \|\nabla_{\Theta_t} f(\boldsymbol{x}^{j_i}, \Theta_t)^\top (\sum_{j=1}^N \nabla_{\Theta_t} f(\boldsymbol{x}_j, \Theta_t)(f(\boldsymbol{x}_j, \Theta_t) - y_j) - p \sum_{j=1}^n \nabla_{\Theta_t} f(\boldsymbol{x}^j, \Theta_t)(f(\boldsymbol{x}^j) - y^j))\|$$

$$\le \|\nabla_{\Theta_t} f(\boldsymbol{x}^{j_i}, \Theta_t)^\top\| \cdot \sum_{j=1}^p \sum_{i=1}^n \|\nabla_{\Theta_t} f(\boldsymbol{x}^j, \Theta_t)(f(\boldsymbol{x}^j, \Theta_t) - y^j) - \nabla_{\Theta_t} f(\boldsymbol{x}_i^j, \Theta_t)(f(\boldsymbol{x}_i^j, \Theta_t) - y_i^j)\|$$

$$(19)$$

$$\le \|\nabla_{\Theta_t} f(\boldsymbol{x}^{j_i}, \Theta_t)^\top\| \cdot \sum_{j=1}^p \sum_{i=1}^n \epsilon(|f(\boldsymbol{x}^j, \Theta_t) - y^j| + \|\nabla_{\Theta_t} f(\boldsymbol{x}_i^j, \Theta_t)\|)$$

$$\le \epsilon \|\nabla_{\Theta_t} f(\boldsymbol{x}^{j_i}, \Theta_t)^\top\| (n^{3/2} R_0 + \frac{\kappa}{\sqrt{m}}) \le \epsilon(\frac{n^{3/2} R_0 \kappa}{\sqrt{m}} + \frac{\kappa^2}{m}).$$

We define that $\epsilon' = A + B$. By the lemma 4, we have that:

$$f(\boldsymbol{x}_i, \Theta_{t+1}) - f(\boldsymbol{x}_i, \Theta_t) = \nabla_{\Theta_t} f(\boldsymbol{x}^{j_i}, \Theta_t)^{\top} [p \sum_{j=1}^{n} \nabla_{\Theta_t} f(\boldsymbol{x}^j, \Theta_t)(f(\boldsymbol{x}^j) - y^j)] \pm \epsilon' \pm \xi_i'(t),$$

where $|\epsilon'| \leq \epsilon(\frac{\kappa R_0}{\sqrt{m}} + \frac{n^{3/2} R_0 \kappa}{\sqrt{m}} + \frac{\kappa^2}{m}) \leq \epsilon(\frac{(\kappa + n^{3/2}) R_0 + \kappa^2}{\sqrt{m}})$; $|\xi_i'(t)| \leq \frac{\eta \kappa^3 R_0^2}{m^{3/2}}$. Based on this, the following vector form can be derived:

$$\boldsymbol{r}_{t+1}^i = p \tilde{\boldsymbol{K}}_e \boldsymbol{r}_t^e \pm \boldsymbol{\epsilon'} \pm \boldsymbol{\xi'(t)} \tag{20}$$

, where $\boldsymbol{r}_{t+1}^i = (f(\boldsymbol{x}_i^j, \Theta_t) - y_i^j)_{j=1}^m, \boldsymbol{r}_t^e = (f(\boldsymbol{x}^j, \Theta_t) - y^j)$. $\qquad \square$

Further, from the derived expression equation 19, we observe that for different $\boldsymbol{r}_{t+1}^i$ we can replace $\boldsymbol{r}_t^e$ with $\boldsymbol{r}_t^i$ as the the pairwise cancellation of some error terms.

## B    ABLATION STUDIES OF SAMPLING STRATEGY

As discussed in Sec. 3.3 , we conduct ablation studies about the effect of sampling points and the size $p$ of sampling group. All experiments are conducted on the first three images of Kodak 24 dataset and the average PSNR values are reported for reference.

### B.1    THE EFFECT OF SAMPLING POINTS

In this subsection, we examine sampling point selection. We randomly sample one point from each group in each iteration, referring to the resulting MLPs as randomly sampling (RI). Additionally, we implement a variant where random sampling occurs only at the beginning, denoting these MLPs as randomly sampling initially (RSI). We also introduce sampling based on the largest residual, termed SLR. Table 4 presents the average PSNR results while remain the same settings as in Experiment 6.1, demonstrating that our method significantly improves various sampling strategies, with SLR yielding the best average performance. Consequently, we adopt SLR for sampling strategy.

Table 4: Ablation experiments on the effect of sampling points. SLR denotes the sampling based on the largest residual in each iteration. RSI denotes the random sampling initially. RI denotes the random sampling in each iteration.

| Method | ReLU+IGA | PE+IGA | SIREN+IGA | Average |
|---|---|---|---|---|
| SLR | 25.44 | 34.09 | 34.85 | 31.46 |
| RSI | 25.52 | 33.77 | 33.76 | 31.02 |
| RI | 25.19 | 34.05 | 33.69 | 30.98 |
| Vanilla | 24.09 | 30.46 | 33.70 | 29.42 |

### B.2    SAMPLING GROUP AND TRAINING TIME

The size $p$ of the sampling group influences the level of data similarity within the group and hardware requirements. Intuitively, a smaller sampling group size reduces errors in estimating training dynamics and allows for more accurate tailored impacts on spectral bias, leading to improved representation performance. However, this leads to a larger matrix $\tilde{\boldsymbol{K}}$, which results in longer run times and greater memory usage.

To further explore the effect of different sizes of sampling groups, we follow the MLP architecture in Experiment 6.1. All training settings are remained. We vary the size from $16 \times 16$ to $128 \times 128$. Larger sizes correspond to coarser estimates of the dynamics. We report the average training time of each iteration.

As shown in Table 5, our method under all sampling sizes has the significant improvement comparing to the baseline models. Particularly, when the size is $128 \times 128$ that means that the size of $\boldsymbol{K}$ is only $24 \times 24$, it still improves the representation performance. That means that relatively

low-precision estimates are sufficient to achieve representation improvements. Further, as the sampling interval decreases, representation accuracy improves; however, the extent of this improvement gradually diminishes with smaller intervals. Specifically, reducing the sampling size from $64^2$ to $32^2$ nearly doubles the average increment $\Delta_{\mathrm{avg}}$. In contrast, the increment becomes negligible when reducing from $32^2$ to $16^2$, while the time required for $16^2$ increases fivefold. This indicates that, in practical applications, there is no need to consider more precise sampling intervals; a relatively coarse partition suffices to achieve substantial improvements while maintaining efficiency.

Table 5: Ablation experiments on sampling group size $p$. $\Delta_{\mathrm{avg}}$ denotes the average increment across three model architectures.

| $p$ | ReLU+IGA | PE+IGA | SIREN+IGA | $\Delta_{\mathrm{avg}}$ | Time (ms) |
|---|---|---|---|---|---|
| $16^2$ | 25.49 | 34.25 | 34.87 | +2.12 | 256.51 |
| $32^2$ | 25.44 | 34.09 | 34.85 | +2.04 | 56.33 |
| $64^2$ | 25.19 | 32.54 | 34.10 | +1.19 | 39.76 |
| $128^2$ | 25.06 | 32.08 | 33.75 | +0.88 | 39.61 |
| Vanilla | 24.09 | 30.46 | 33.70 | - | 30.00 |

## C EMPIRICAL ANALYSIS ON SIMPLE FUNCTION APPROXIMATION

In this section, we first visualize more experimental results **Experiment 1** in our paper. Subsequently, we examine impacts on spectral bias of ReLU and SIREN with varying numbers of balanced eigenvalues under different group sizes from **Experiment 2**.

As can been seen in Fig. 7, the same trend as our observations in our paper, with the width increases, the differences on MSE curves and spectrum between gradient adjustments by $\boldsymbol{S}$, $\tilde{\boldsymbol{S}}$ and $\tilde{\boldsymbol{S}}_e$ gradually diminish when $end = 13, 15$. This is consistent with our Theorem 4.1 and the analysis of our Theorem 4.2. As shown in Fig. 7, 8 and 9, impacts on spectral bias are gradually amplified by increasing the number of balanced eigenvalues of $\tilde{\boldsymbol{S}}_e$, which is also consistent with the observations in our paper. The above empirical results corroborate our theoretical results and illustrate that how our inductive gradient adjustment method tailor impacts on spectral bias.

## D PER-SCENE RESULTS OF 2D COLOR IMAGE APPROXIMATION

In this section, we provide metrics of each image and give more visualization. Peak signal to noise ratio (PSNR) of 2D color image approximation results are shown in Table 6. Structural Similarity Index Measure (SSIM) of 2D color image approximation results are shown in Table 7. Learned Perceptual Image Patch Similarity (LPIPS) by the 'alex' in Zhang et al. (2018) of 2D color image approximation results are also reported in Table 8. We visualize the representation results of all methods on Kodak 5 and Kodak 6 in Fig. 10.

As can be seen in Table 6, 7 and 8, our IGA method achieve the best average results in three metrics across three model architectures. And better representation results can be found in Fig. 10.

## E EXPERIMENTAL DETAILS AND PER-SCENE RESULTS OF 3D SHAPE REPRESENTATION

In this section, we firstly introduce the detailed training strategy. Then we provide the intersection over union (IOU) metric value of each 3D shape and give more visualization.

Following previous works (Saragadam et al., 2023; Shi et al., 2024a; Cai et al., 2024), we train all models for 200 epochs with a learning rate decay exponentially to $0.1$ of initial rates. We set the initial learning rate as $2e - 3$ for ReLU and PE; For SIREN, we set the initial learning rate as $5e - 4$. FR, BN and our IGA method are adopted the same learning rate with baseline models. We set $end = 7$ for ReLU and PE; For SIREN, we set $end = 14$. The per-scene results are reported in Table 11. It can be observed that our IGA method achieves improvements in IOU metrics across all

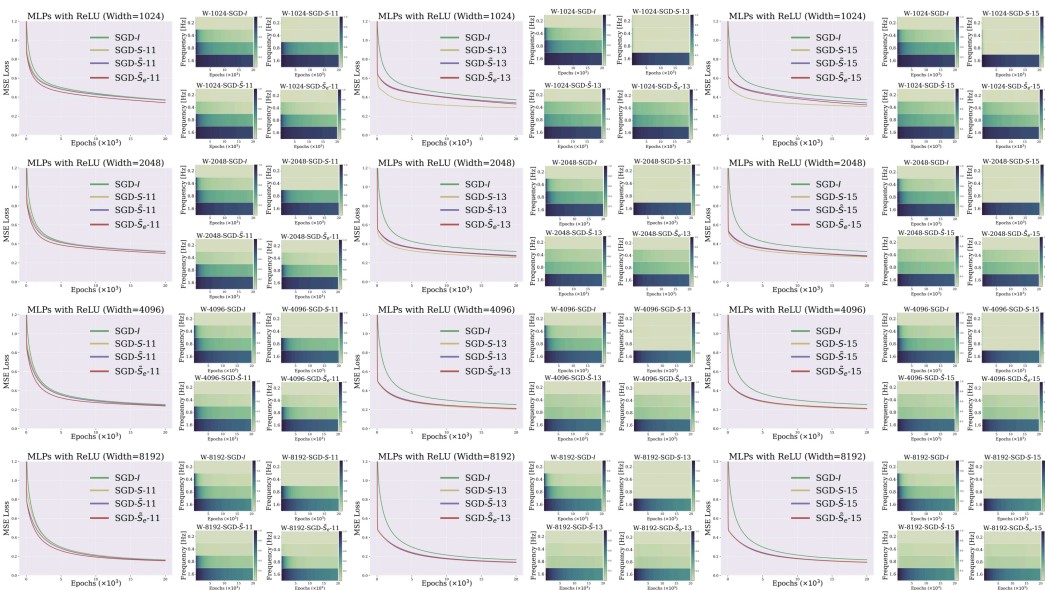

Figure 6: Evolution of approximation error with training iterations on time domain and Fourier domain. Line plots visualize the MSE loss curves of MLPs. Heatmaps show the relative error $\Delta_k$ on four frequency bands, where the frequency indices are labeled in ascending order based on their frequency values. SGD-$\tilde{\boldsymbol{S}}_e$-11 denotes that the MLP with ReLU optimized using gradients adjusted by $\tilde{\boldsymbol{S}}_e$ with $end = 11$ and group size $p = 8$.

Table 6: Peak signal to noise ratio (PSNR ↑) of 2D color image approximation results by different methods. The detailed settings can be found in Sec. 6.1.

| Method | Kodak1 | Kodak2 | Kodak3 | Kodak4 | Kodak5 | Kodak6 | Kodak7 | Kodak8 | Average |
|---|---|---|---|---|---|---|---|---|---|
| ReLU | 19.78 | 26.62 | 25.88 | 24.70 | 17.70 | 21.98 | 21.52 | 16.07 | 21.78 |
| ReLU+FR | 19.96 | 26.63 | 26.53 | 25.31 | 18.05 | 22.03 | 22.13 | 16.52 | 22.14 |
| ReLU+BN | 20.21 | 27.24 | 27.24 | 25.52 | 18.55 | 22.26 | 22.66 | 16.75 | 22.55 |
| **ReLU+IGA** | 20.42 | 27.71 | 28.20 | 26.17 | 18.81 | 22.69 | 22.96 | 16.99 | **23.00** |
| PE | 26.07 | 32.51 | 32.80 | 31.37 | 24.60 | 27.28 | 31.74 | 22.79 | 28.64 |
| PE+FR | 26.95 | 32.23 | 33.92 | 32.14 | 26.63 | 28.49 | 32.71 | 24.85 | 29.74 |
| PE+BN | 29.22 | 35.03 | 35.67 | 33.96 | 27.42 | 30.03 | 35.54 | 26.36 | 31.65 |
| **PE+IGA** | 29.17 | 35.18 | 37.91 | 35.07 | 28.06 | 31.27 | 36.60 | 26.41 | **32.46** |
| SIREN | 29.61 | 35.19 | 36.31 | 35.10 | 29.74 | 31.01 | 36.73 | 27.50 | 32.65 |
| SIREN+FR | 30.00 | 34.73 | 36.95 | 34.87 | 29.72 | 30.79 | 36.31 | 27.52 | 32.61 |
| SIREN+BN | 29.11 | 34.75 | 36.86 | 34.74 | 29.33 | 30.41 | 36.53 | 27.08 | 32.35 |
| **SIREN+IGA** | 30.10 | 35.85 | 38.60 | 35.80 | 30.13 | 31.94 | 37.68 | 27.73 | **33.48** |

objects, and achieves the best performance in average metrics of all objects. We visualize the Thai and Lucy objects in Fig. 11.

# F  EXPERIMENTAL DETAILS AND PER-SCENE RESULTS OF LEARNING NEURAL RADIANCE FIELDS

In this section, we firstly introduce some training details. Then we provide metrics of each scenario and give more visualization. As previously discussed in Experiment 6.3, we apply our method on the original NeRF model (Mildenhall et al., 2021). The original NeRF model adopts two models: "coarse" and "fine" models. We adjust the gradients of the "fine" model as it captures more high-frequency details and remain gradients of the "coarse" model. The "NeRF-pytorch" codebase (Yen-Chen, 2020) is used and we follow its default settings. Besides, we also compare to previous training dynamics methods, i.e., FR and BN. Their hyperparaemters follows their publicly codes. Generally, we set the $end = 25$. For more complex scenes, such as the branches and leaves of the ficus, we set

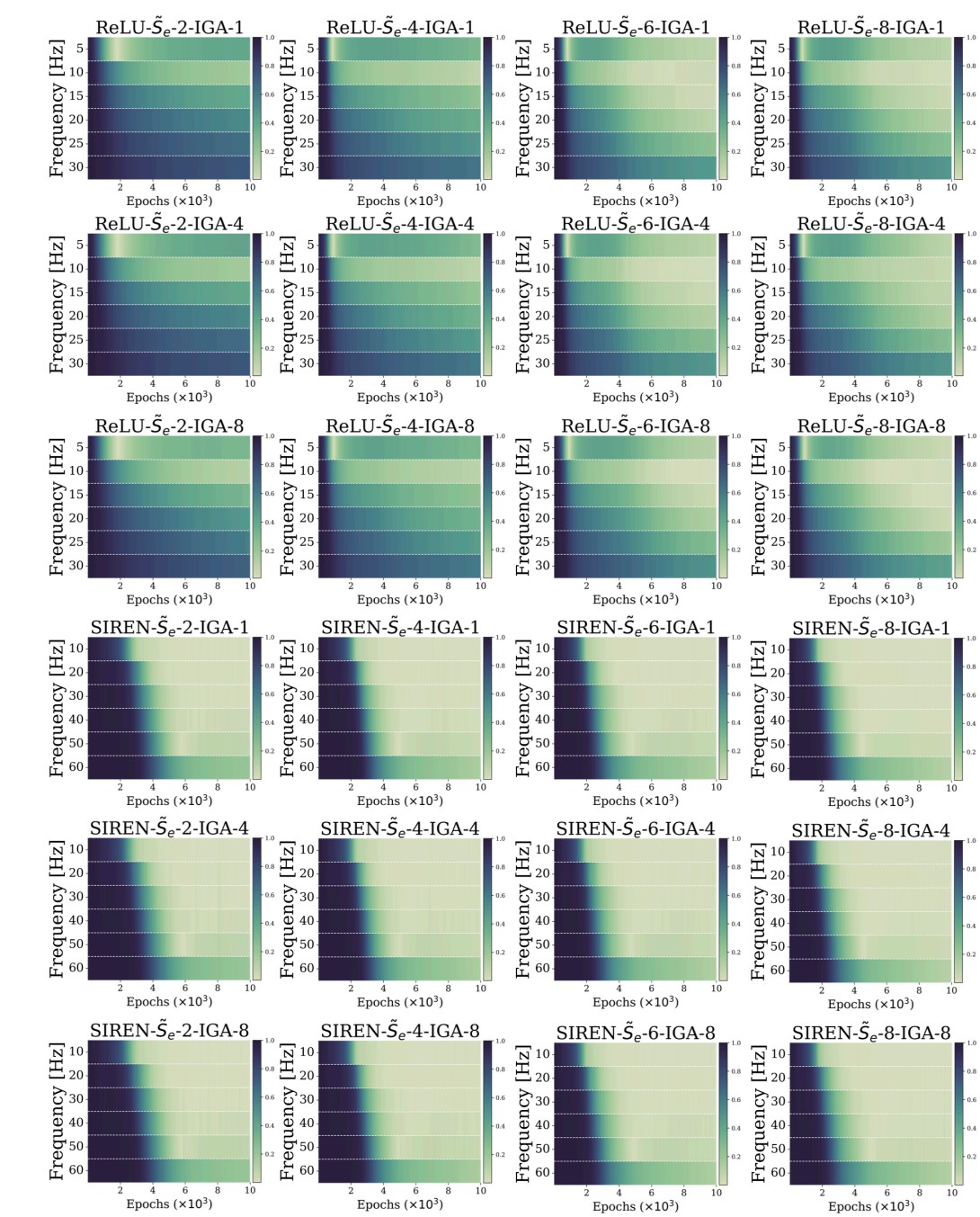

Figure 7: Progressively amplified impacts on spectral bias of ReLU and SIREN by increasing the number of balanced eigenvalues of $\tilde{\boldsymbol{S}}_e$ when the group size is $8$. ReLU denotes that the MLP with ReLU optimized using vanilla gradients; ReLU-$\tilde{\boldsymbol{S}}_e$-2-IGA-8 denotes that the MLP with ReLU optimized using gradients adjusted by $\tilde{\boldsymbol{S}}_e$ with $end = 2$ and group size is $8$. Results of other numbers of balanced eigenvalues can be found in the Appendix.

the $end = 30$ to achieve better learning of high frequencies. We report per-scene results in Table 10 and visualize several scenes in Fig. 12.

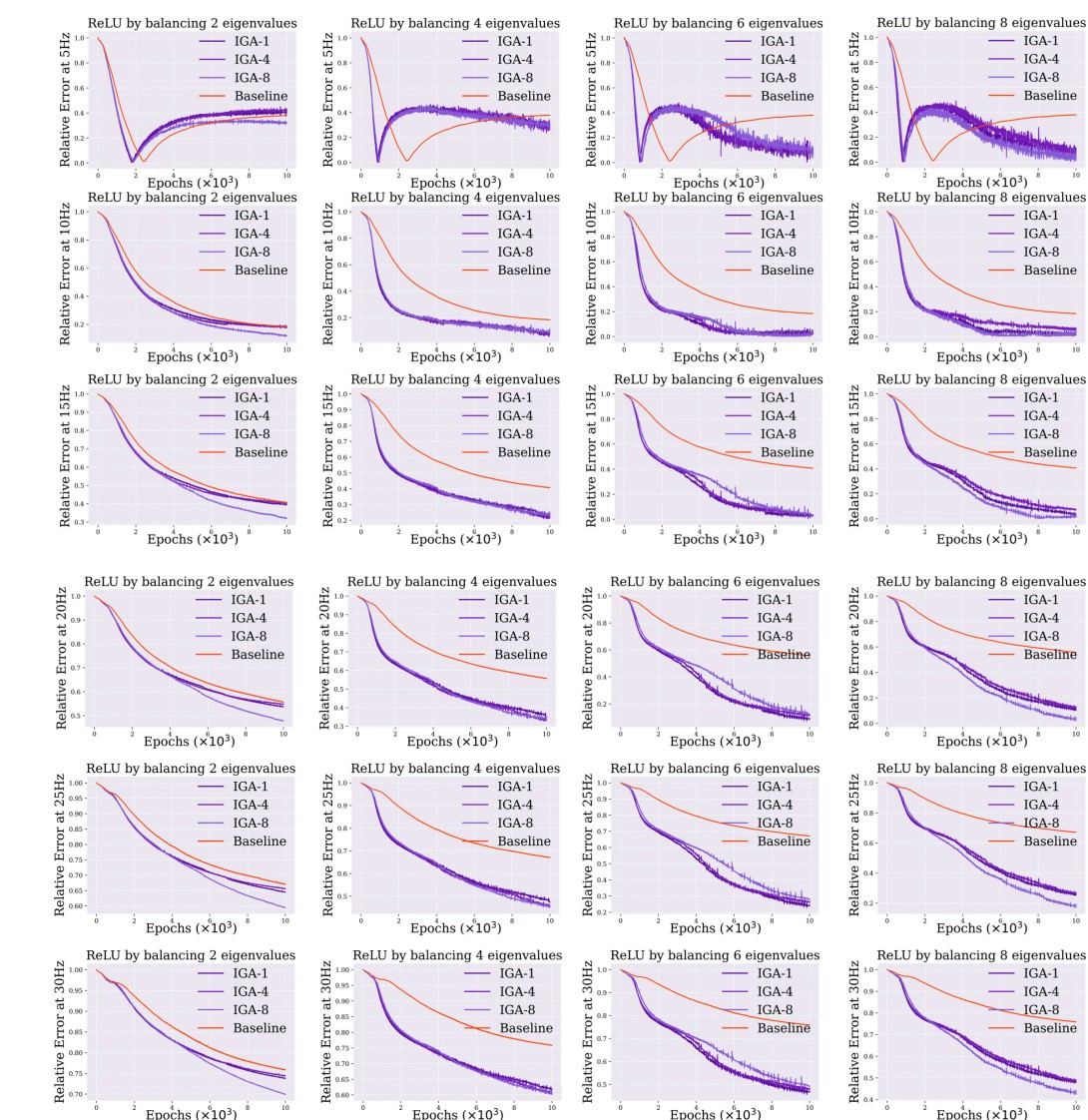

Figure 8: Comparison of group sizes with varying balanced eigenvalues on Relative Error at 5, 10, 15, 20, 25, 30Hz of ReLU. IGA-1 denotes that the group size is 1, i.e., the eNTK-based gradient adjustment; IGA-4 denotes that the group size is 4 for IGA; Baseline denotes MLPs with ReLU activation function optimized by vanilla gradients.

Table 7: Structural Similarity Index Measure (SSIM ↑) of 2D color image approximation results by different methods. The detailed settings can be found in Sec. 6.1.

| Method | Kodak1 | Kodak2 | Kodak3 | Kodak4 | Kodak5 | Kodak6 | Kodak7 | Kodak8 | Average |
|---|---|---|---|---|---|---|---|---|---|
| ReLU | 0.2978 | 0.6455 | 0.7235 | 0.6199 | 0.2797 | 0.4531 | 0.5681 | 0.2785 | 0.4833 |
| ReLU+FR | 0.3063 | 0.6450 | 0.7310 | 0.6302 | 0.2920 | 0.4546 | 0.5789 | 0.2972 | 0.4919 |
| ReLU+BN | 0.3177 | 0.6548 | 0.7426 | 0.6375 | 0.3164 | 0.4660 | 0.5979 | 0.3044 | 0.5047 |
| **ReLU+IGA** | 0.3298 | 0.6628 | 0.7564 | 0.6387 | 0.3195 | 0.4753 | 0.5946 | 0.3232 | **0.5126** |
| PE | 0.7193 | 0.8126 | 0.8634 | 0.8077 | 0.7371 | 0.7553 | 0.8911 | 0.6790 | 0.7832 |
| PE+FR | 0.7653 | 0.8082 | 0.8819 | 0.8266 | 0.8074 | 0.7953 | 0.8947 | 0.7546 | 0.8167 |
| PE+BN | 0.8473 | 0.8803 | 0.9137 | 0.8783 | 0.8395 | 0.8531 | 0.9372 | 0.8097 | 0.8699 |
| **PE+IGA** | 0.8458 | 0.8806 | 0.9402 | 0.8994 | 0.8542 | 0.8778 | 0.9462 | 0.8132 | **0.8822** |
| SIREN | 0.8715 | 0.8948 | 0.9168 | 0.9060 | 0.9040 | 0.8689 | 0.9534 | 0.8643 | 0.8975 |
| SIREN+FR | 0.8781 | 0.8869 | 0.9308 | 0.9011 | 0.8971 | 0.8797 | 0.9536 | 0.8654 | 0.8991 |
| SIREN+BN | 0.8595 | 0.8818 | 0.9366 | 0.9002 | 0.8958 | 0.8661 | 0.9545 | 0.8552 | 0.8937 |
| **SIREN+TDS** | 0.8830 | 0.9070 | 0.9492 | 0.9173 | 0.9110 | 0.8996 | 0.9606 | 0.8688 | **0.9121** |

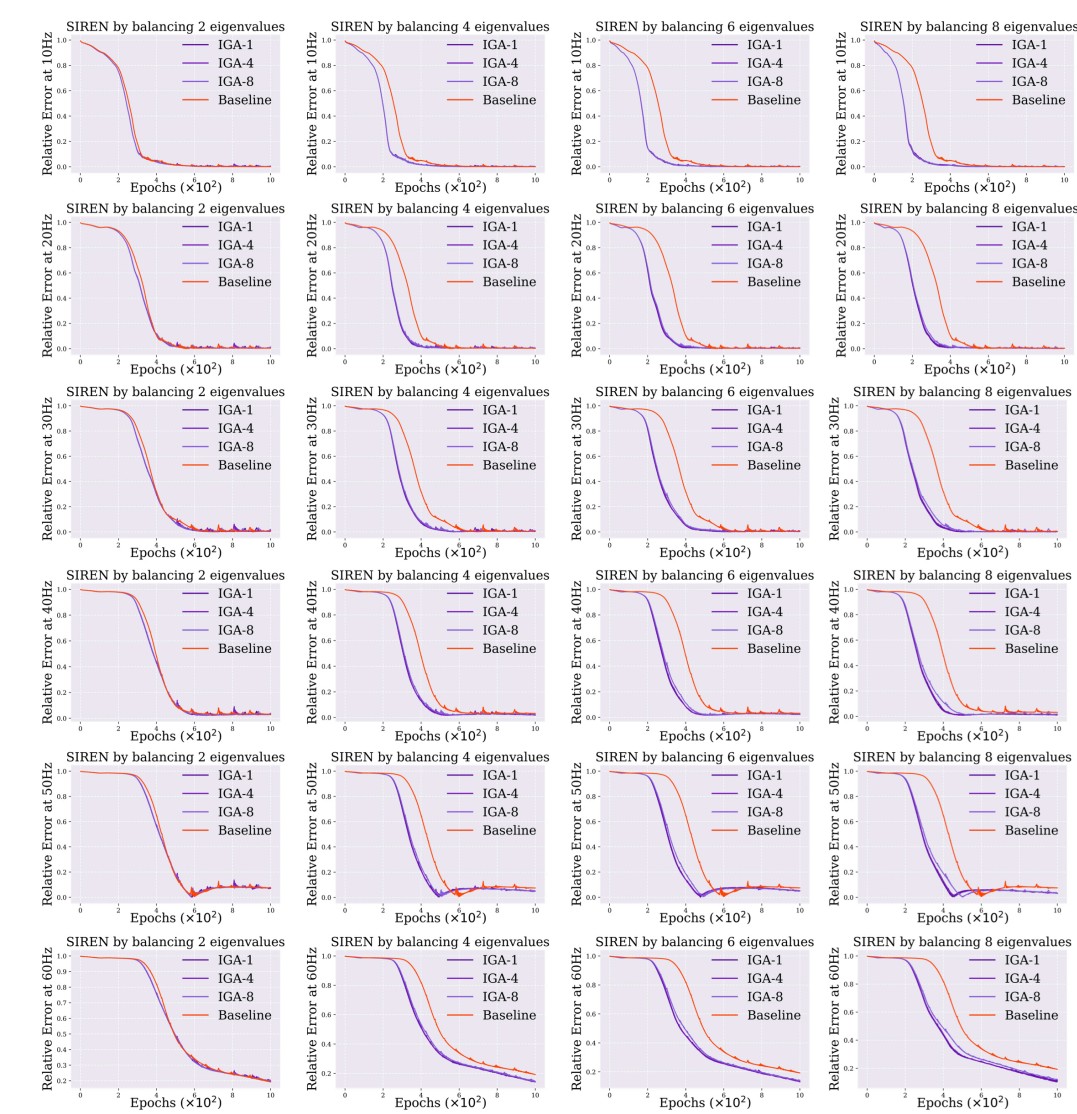

Figure 9: Comparison of group sizes with varying balanced eigenvalues on Relative Error at 10, 20, 30, 40, 50, 60Hz of SIREN. IGA-1 denotes that the group size is 1, i.e., the eNTK-based gradient adjustment; IGA-4 denotes that the group size is 4 for IGA; Baseline denotes the SIREN optimized by vanilla gradients.

Table 8: Learned Perceptual Image Patch Similarity (LPIPS ↓) of 2D color image approximation results by different methods. The detailed settings can be found in Sec. 6.1.

| Method | Kodak1 | Kodak2 | Kodak3 | Kodak4 | Kodak5 | Kodak6 | Kodak7 | Kodak8 | Average |
|---|---|---|---|---|---|---|---|---|---|
| ReLU | 0.7571 | 0.5392 | 0.4000 | 0.5547 | 0.7788 | 0.5854 | 0.6172 | 0.8089 | 0.6302 |
| ReLU+FR | 0.7692 | 0.5513 | 0.3871 | 0.5362 | 0.7829 | 0.6110 | 0.6131 | 0.8008 | 0.6315 |
| ReLU+BN | 0.7332 | 0.5230 | 0.3530 | 0.4900 | 0.7115 | 0.5956 | 0.5652 | 0.7849 | 0.5945 |
| **ReLU+IGA** | 0.6857 | 0.4770 | 0.2927 | 0.4730 | 0.6695 | 0.5609 | 0.5278 | 0.7527 | **0.5549** |
| PE | 0.2803 | 0.2092 | 0.1062 | 0.2128 | 0.2643 | 0.2517 | 0.1218 | 0.3322 | 0.2223 |
| PE+FR | 0.2547 | 0.2278 | 0.0905 | 0.1897 | 0.1566 | 0.2179 | 0.1154 | 0.2426 | 0.1869 |
| PE+BN | 0.1378 | 0.1023 | 0.0545 | 0.1257 | 0.1435 | 0.1430 | 0.0455 | 0.1579 | 0.1138 |
| **PE+IGA** | 0.1363 | 0.0938 | 0.0256 | 0.0956 | 0.1056 | 0.1068 | 0.0310 | 0.1559 | **0.0938** |
| SIREN | 0.1052 | 0.0878 | 0.0661 | 0.0994 | 0.0654 | 0.0967 | 0.0271 | 0.0980 | 0.0807 |
| SIREN+FR | 0.0928 | 0.0888 | 0.0465 | 0.1121 | 0.0661 | 0.1258 | 0.0317 | 0.0865 | 0.0813 |
| SIREN+BN | 0.1289 | 0.1338 | 0.0424 | 0.1242 | 0.0847 | 0.1407 | 0.0313 | 0.1268 | 0.1016 |
| **SIREN+IGA** | 0.0895 | 0.0746 | 0.0246 | 0.0858 | 0.0628 | 0.0853 | 0.0199 | 0.0921 | **0.0668** |

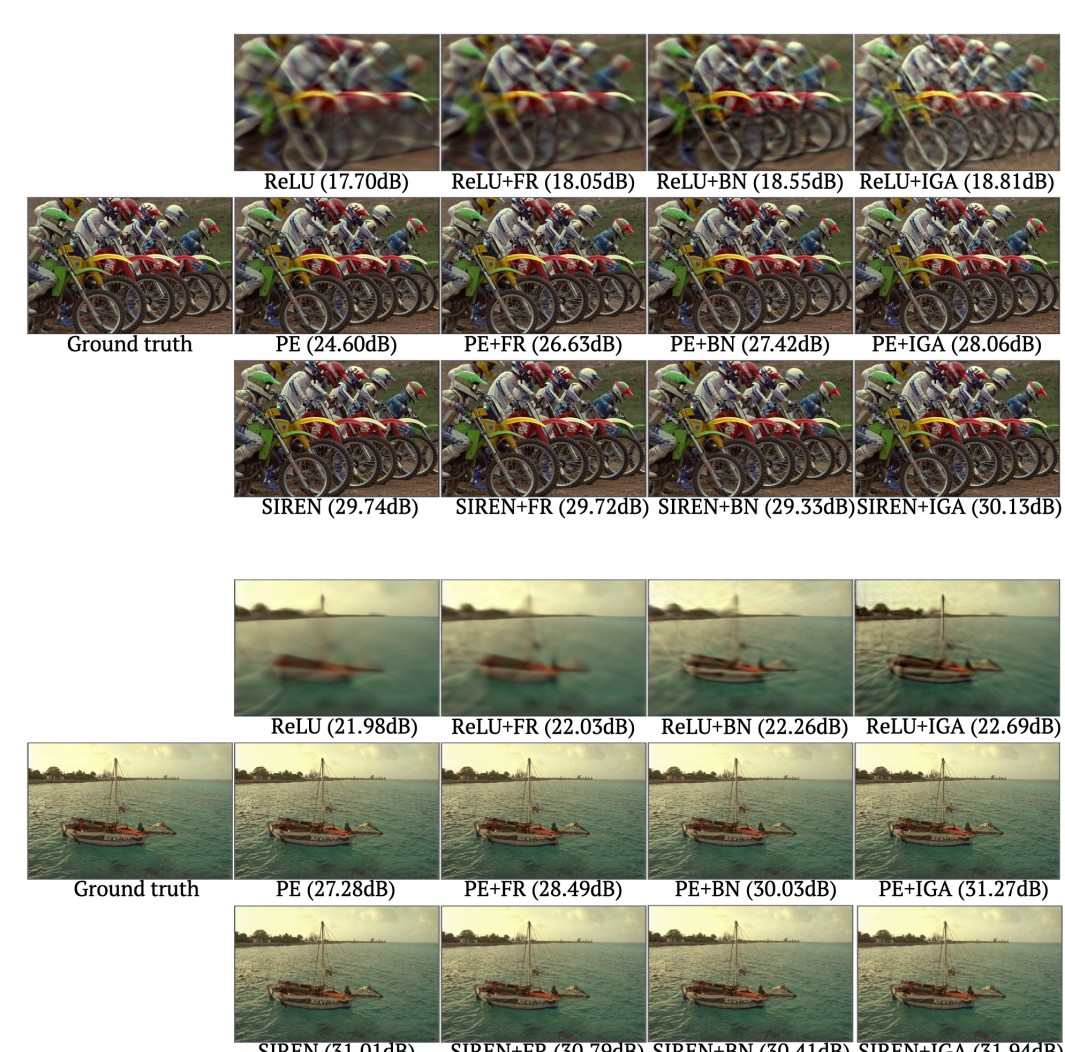

Figure 10: Visualization of 2D color image approximation results and PSNR values on Kodak 5 (top) and Kodak 6 (bottom). ReLU+IGA denotes that the MLPs with ReLU activation function are optimized by our IGA method.

Table 9: Intersection over Union (IOU) of 3D shape representation by different methods. The detailed settings can be found in Sec. 6.2 and Appendix E.

| Method | Thai | Armadillo | Dragon | Bun | Lucy | Average |
|---|---|---|---|---|---|---|
| ReLU | 0.9379 | 0.9756 | 0.9708 | 0.9924 | 0.9467 | 0.9647 |
| ReLU+FR | 0.9428 | 0.9756 | 0.9707 | 0.9914 | 0.9465 | 0.9654 |
| ReLU+BN | 0.9260 | 0.9677 | 0.9612 | 0.9873 | 0.9286 | 0.9542 |
| **ReLU+IGA** | 0.9563 | 0.9805 | 0.9793 | 0.9931 | 0.9571 | **0.9733** |
| PE | 0.9897 | 0.9956 | 0.9953 | 0.9985 | 0.9920 | 0.9942 |
| PE+FR | 0.9929 | 0.9979 | 0.9964 | 0.9990 | 0.9945 | 0.9961 |
| PE+BN | 0.9912 | 0.9960 | 0.9941 | 0.9987 | 0.9892 | 0.9938 |
| **PE+IGA** | 0.9943 | 0.9979 | 0.9975 | 0.9990 | 0.9961 | **0.9970** |
| SIREN | 0.9786 | 0.9908 | 0.9937 | 0.9953 | 0.9862 | 0.9889 |
| SIREN+FR | 0.9731 | 0.9935 | 0.9905 | 0.9972 | 0.9785 | 0.9866 |
| SIREN+BN | 0.9788 | 0.9931 | 0.9612 | 0.9970 | 0.9823 | 0.9825 |
| **SIREN+IGA** | 0.9802 | 0.9920 | 0.9938 | 0.9958 | 0.9868 | **0.9897** |

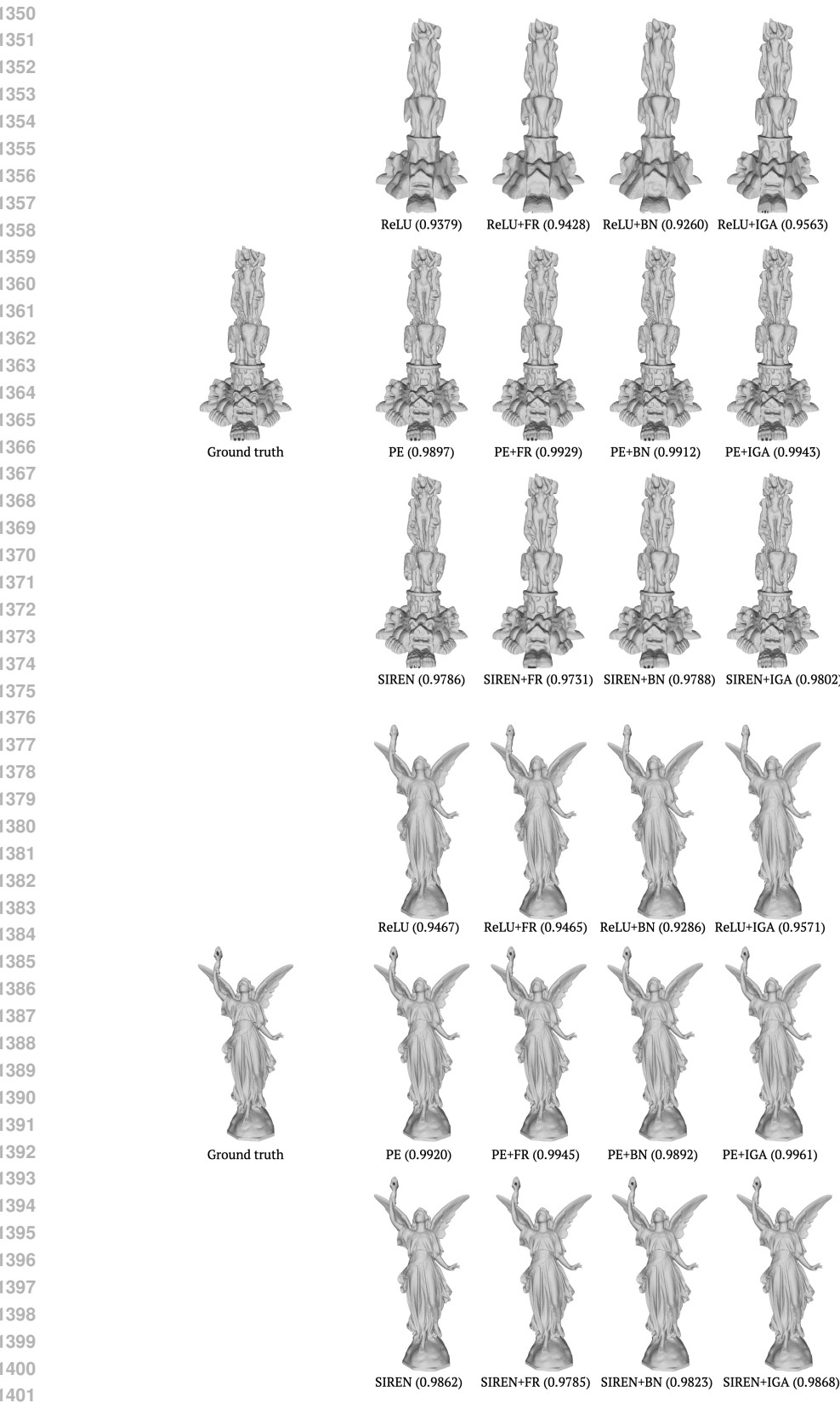

Figure 11: Visualization of 3D shape representation results and IOU values on the Thai (top) and the Lucy (bottom). ReLU+IGA denotes that the MLPs with ReLU activation function are optimized by our IGA method.

Table 10: Per-scene results of learning 5D neural radiance fields by different methods. Details can be found in Sec. 6.3 and Appendix F.

| | Methods | Ship | Materials | Chair | Ficus | Hotdog | Drums | Mic | Lego | Average |
|---|---|---|---|---|---|---|---|---|---|---|
| PSNR ← | NeRF | 29.30 | 29.55 | 34.52 | 29.14 | 36.78 | 25.66 | 33.37 | 31.53 | 31.23 |
| | NeRF+FR | 29.50 | 29.70 | 34.54 | 29.35 | 37.08 | 25.74 | 33.35 | 31.55 | 31.35 |
| | **NeRF+IGA** | 29.49 | 29.87 | 34.69 | 29.38 | 37.22 | 25.80 | 33.48 | 31.83 | **31.47** |
| SSIM ← | NeRF | 0.8693 | 0.9577 | 0.9795 | 0.9647 | 0.9793 | 0.9293 | 0.9783 | 0.9626 | 0.9526 |
| | NeRF+FR | 0.8729 | 0.9600 | 0.9797 | 0.9665 | 0.9792 | 0.9304 | 0.9786 | 0.9626 | 0.9537 |
| | **NeRF+IGA** | 0.8716 | 0.9619 | 0.9807 | 0.9667 | 0.9801 | 0.9315 | 0.9791 | 0.9648 | **0.9546** |
| LPIPS → | NeRF | 0.077 | 0.021 | 0.011 | 0.021 | 0.012 | 0.052 | 0.022 | 0.019 | 0.029 |
| | NeRF+FR | 0.071 | 0.020 | 0.011 | 0.019 | 0.013 | 0.050 | 0.021 | 0.019 | 0.028 |
| | **NeRF+IGA** | 0.074 | 0.019 | 0.010 | 0.019 | 0.011 | 0.049 | 0.020 | 0.017 | **0.027** |

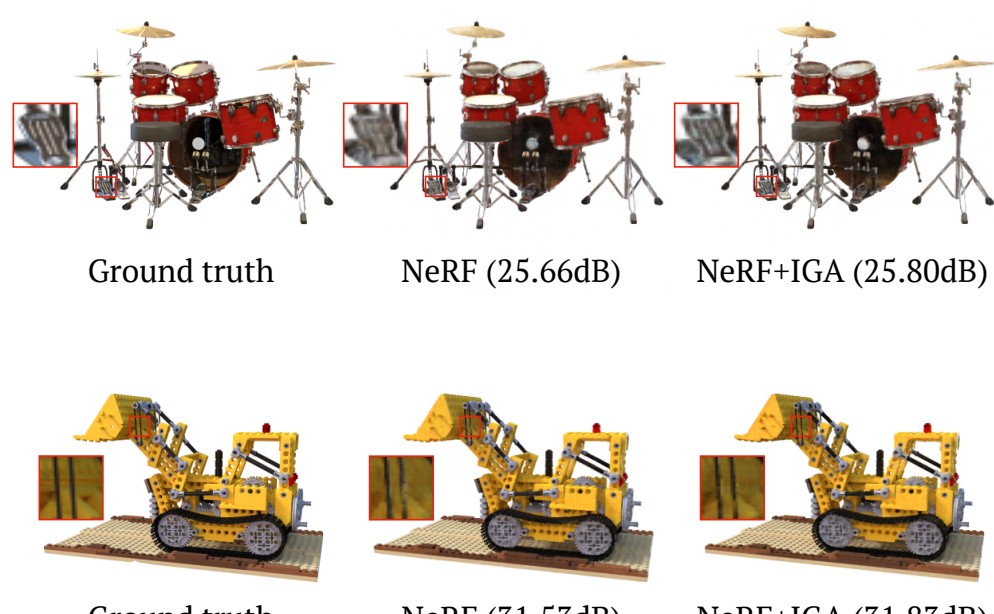

| Ground truth | NeRF (25.66dB) | NeRF+IGA (25.80dB) |
|---|---|---|

| Ground truth | NeRF (31.53dB) | NeRF+IGA (31.83dB) |
|---|---|---|

Figure 12: Visual examples of novel view synthesis results of NeRF and NeRF+IGA.

