# OpenReview forum: "INDUCTIVE GRADIENT ADJUSTMENT FOR SPECTRAL BIAS IN IMPLICIT NEURAL REPRESENTATIONS"
_ICLR.cc/2025/Conference — ICLR 2025 Conference Withdrawn Submission_

### Official Review · Reviewer_SnNW · 2024-11-02

**Soundness:** 4
**Presentation:** 4
**Contribution:** 3
**Rating:** 8
**Confidence:** 4

**Summary:**

This paper proposes an inductive gradient adjustment technique for INRs. By analyzing the empirical NTK, the authors link the spectral bias in multi-layer perceptrons (MLPs) to training dynamics. Using the eNTK-based gradient transformation matrix, they adjust gradients to reduce spectral bias, resulting in improved generalization. Tested across different INR tasks and architectures, this approach yields better texture details and sharper edges, outperforming current training methods.

**Strengths:**

The paper's strengths are its theoretical and practical advancements in mitigating spectral bias in MLPs for INRs. Specifically:

1  Inspired by the established connection between spectral bias and MLP dynamics in previous works, the authors propose a theoretically guided training adjustment strategy. This method is simple yet direct and effective.

2 Theorem 4.2 is novel and extends NTK theory by considering the number of sampled data points. It has practical value by incorporating two key factors: data similarity and neural network width. Its setting is well-suited for INRs. Moreover, Experiments 1 and 2 are impressive and help readers grasp the practical significance of Theorems 4.1 and 4.2.

3 Comprehensive experiments on various INR tasks confirm that this method yields more high-frequency details and outperforms existing approaches in controlling spectral bias for sharper and more detailed representations.

**Weaknesses:**

1 The effectiveness of the proposed method depends on the pre-selection of the hyperparameters "start" and "end."  However, it seems that the stragety of hyperparameter choosing is not discussed, and the corresponding ablation studies are missing.

2 The motivation of Theorem 4.2 is unclear before reading Experiment 2. I recommend that the authors add some breif discussion on the motivation of Theorem 4.2 (like lines 184-186) before presenting it.

3 Theorem 4.2 demonstrates that the proposed method is effective when data in each group is similar. How to improve the method when data similarity is low? Moreover, how could this method be extended to the case of non-uniform sampling?  Data similarity may vary significantly across groups, suggesting a need for a  non-uniform sampling strategy.

**Questions:**

In lines 289-294, what does "this error" mean?

---

### Official Review · Reviewer_VijA · 2024-11-03

**Soundness:** 3
**Presentation:** 1
**Contribution:** 3
**Rating:** 3
**Confidence:** 2

**Summary:**

This work describes how one can use the NTK framework to alter the vanilla gradient descent of Implicit Neural Representations (INRs) to better match the expected spectrum of the target signal.

**Strengths:**

- I liked that the main message of the theorems was states in plain english.
- Tackling practical applications is essential for the progress of our field.
- Experiments are carried out in a very diverse setting.

**Weaknesses:**

My assessment is mainly based on the poor framing of this paper with respect to the paper by Geifman et al. 2023.

- abstract: The sentence "The superiority representation performance clearly validate the advantage of our proposed method." not only has typos (these can be corrected using LLMs), but it's also very vague and non quantitative: how superior? what is representation performance?
- Motivation: "Moreover, there is no clear guidance on the choice of Fourier bases matrix for reparameterization or batch normalization layer to confront spectral bias with varying degrees in a variety of INR tasks."; while I understand in principle how that can be a problem, I don't recall a precise discussion in the literature of how this problem can affect users of INRs. I think it would be nice to discuss how sensitive INRs are to bad choices of these 2 ingredients.
- Scalar outputs: I think it could be nice to discuss in the methods section why we chose in the theoretical analysis to focus on scalar signals, and how limiting that can be.
- Clarity: while there are a lot of typos that don't help with clarity, there is a more profound problem in my opinion. The section 3. is a great example in that there are a lot of operations that are defined vaguely or with hand-wavy terms making the paper very difficult to follow. For example, $f$ is sometimes applied to a sample $x_i$, sometimes to the full dataset $X$. $S$ is introduced in the gradient descent algorithm without clear motivation and it's not clear until we read it afterwards that it was designed by Geifman et al. It's unclear why $\tilde{K}$ is empirical: it's rather dynamic rather fixed no? As a consequence the dependency on $t$ should be marked somehow. The division of samples into groups is not explicited. I guess it's really some sort of clustering that's happening.
- Theorem: the assumptions should be laid out clearly before stating the theorem. The theorem needs to be cleaned up to feature way less notations: currently it's very difficult to parse through it. I think an informal sketch of the proof could be given in the main text.
- Relationship to Geifman et al. 2023: I think it's very unclear when reading the paper, that the core idea of modifying gradient descent using a matrix derived from the NTK originated from Geifman et al. 2023. This is further obfuscated by not using the same denominations as in this predating paper, and resorting to a non-motivated name for the method. I think it should be clear as soon as reading the abstract that this paper presents a way to apply this original method in practice, e.g. to large datasets.
I think it should also be clear in the large scale experiments how difficult it would be to apply directly the method proposed by Geifman et al.

**Questions:**

- "projection directions related to low frequencies": what type of frequencies are we talking about here?
- In table 3. it seems that IDA is replaced by TDS (not introduced beforehand): what is TDS?
- "IGA enables NeRF to capture more accurate and high-frequency reconstruction results, thereby improving PSNR values." : I thought that PSNR, as an MSE-based metric, was less sensitive to high frequency reconstruction, so I am not sure I understand the point, wouldn't a multiscale-SSIM be more appropriate to judge quantitatively of the reconstruction of high frequency details?

**Details Of Ethics Concerns:**

I think this paper does a bad job of giving credit to Geifman et al. 2023 "Controlling the Inductive Bias of Wide Neural Networks by Modifying the Kernel’s Spectrum" that invented the core method of modifying gradient descent by multiplying the jacobian by an adhoc matrix and rather tries to obfuscate the fact that this core contribution is not novel by for example using a different name.

I do want to clarify that I think that the paper has an interesting content when it comes to the practical application of the method, but the core theoretical idea is not clearly presented as originated from a different paper.

---

### Official Review · Reviewer_NZyw · 2024-11-04

**Soundness:** 3
**Presentation:** 2
**Contribution:** 2
**Rating:** 5
**Confidence:** 3

**Summary:**

The paper titled " Inductive Gradient Adjustment for Spectral Bias in Implicit Neural Networks", explores the challenges of spectral bias in MLPs. It delves into the linear dynamics model of MLPs to identify the empirical Neural Tangent Kernel (eNTK) matrix. It then proposes a practical inductive gradient adjustment method to  improve the spectral bias through inductive generalization of the eNTK-based gradient transformation matrix.

**Strengths:**

The strengths of this paper are as follows:
1) The idea is based on a solid foundation, that is to build a customized eNTK matrix that will facilitate convergence and also reduce spectral bias. The connection of NTK with spectal bias and the performance of INR has been shown by Tancik. This paper attempts to adjust the NLTH matrix by designing a eNTK matrix that has more desirable properties. This allows for tailored impacts on spectral bias, enabling more precise implicit neural representations (INRs).
2) The  method is designed to be practical and can work with millions of data points, making it applicable to a wide range of INR tasks.
3) The paper provides extensive experimental analyses across various INR tasks, demonstrating the superiority of the proposed training dynamics adjustment strategy over previous methods.

**Weaknesses:**

1) Even though the proposed motivation is indeed a decent idea, the paper does not offer a precise theoretical justification of how this is the only way to adjust/generate the eNTK matrix. Note that there is no precise mathematical quantification of spectral bias. Therefore it simply not possible to evaluate the quantitative improvements to spectral bias by this approach. Therefore the main outcomes that the paper provides is making the computation of eNTK matrix more tractable by grouping and sampling pixels and then smoothing out the eigen spectrum of the transformation matrix. On its own, neither of these are particularly unique ideas since similar strategies have been used before.
2) The theorems are unremarkable and a bit hard to read. It is obvious that based on sampling, one can show bounds to the original matrix using perturbation theory. The authors should focus more of writing clearly what intuition they are trying to convey, rather than just the mathematical notations.
3) The paper is generally hard to read specially the math notation. At places, notations have not been defined such as S^t_tds in line 200. Else where multiple definition of \tilde{K_e} has been introduced. The authors would benefit from a through proof reading and clarification of math notations.

**Questions:**

na

---

### Official Review · Reviewer_LggB · 2024-11-08

**Soundness:** 3
**Presentation:** 1
**Contribution:** 2
**Rating:** 3
**Confidence:** 3

**Summary:**

This paper considers the training of Implicit Neural Representations (INR) using MLP. Studying the NTK dynamics of the MLP, it is known that the learning function is biased towards lower frequencies in the signal to represent. The authors propose a pre-conditioning scheme that is both computationally tractable and mitigates the spectral bias. This scheme is based on an approximation of the NTK, with the current parameters and a few well-selected data points. Theoretical results show that this approximation should have a similar impact as if the preconditioning was computed on the true NTK. Then, the method is evaluated on simulated data and on three vision tasks, showing improvement relative to the standard technique to reduce this bias.

**Strengths:**

- The experimental section is well furnished and has various applications to real data.
- The theoretical results seem sound.

**Weaknesses:**

- **W1:** The writing is not always clear or rigorous, making the paper hard to follow. In particular, there are many notations that are not defined, and some quantities are changed from one section to another (see various questions below).

- **W2:** A significant part of the methodological contribution (Eq. 3, and Theorem 4.1) overlaps what was proposed by Geifman et al. (2024). A clearer positioning of the contribution relative to this paper is needed.

- **W3:** The practical benefit of the method is not demonstrated by the applications (see **Q7**).

**Questions:**

- **Q1:** In some cases, this spectral bias can be seen as beneficial, as it can reduce the noise that is often present in the high frequencies of the data. In this paper, the authors mostly consider that the data which is fed to the model is noiseless. How would the results change if the data was noisy? Would the spectral bias still be something that we which to avoid in this case?

- **Q2:** The authors mention in l.188 that they sort the data by proximity of input samples. It is not clear how this is done for multi-dimensional data when $d_0 > 1$. Could the authors develop on this point? These looks like some kind of clustering but it is not clear from the manuscript.

- **Q3:** In line 198, the sum is not on $N$ but on $n$ which is supposed to be much smaller than $N$ if we want to improve the computational complexity. However, this means that instead of computing a dynamic in the full parameter space, we are restricted to a subspace of size $n$

- **Q4:** I don't understand what the authors mean in paragraph l.219. The equation (4) should be extended but it is not clear how this is done and how it relates to controlling $\lambda_{end}$.

- **Q5:** The assumption on the uniform convergence of the model on each group with $|f(x_{i_1}) - y_1 - f(x_{i_2}) + y_2| < \epsilon$ and $\|\nabla_\Theta  f (x_{i_1} ) − \nabla_\Theta f (x_{i_2} )\| < \epsilon$ seems quite strong. Don't they clash with the idea that the data contains high frequencies? how plausible are they?

- **Q6:** The authors do not discuss the varying computational cost of the method. In particular, for the proposed method compared to classical Adam, a preconditioner needs to be computed at each iteration. How costly it is? Could the results with $S = I$ and $S = S_e$ be compared relative to time?

- **Q7**: I am not an expert on the INR, but I have issues understanding the purpose of the vision application, which appears to lack any consideration for the generalization of the learned model. From what I understand, in the three applications, a single image/volume/radiance field is used to fit an INR model and the evaluation of the model is based on the same points. However, this does not look like realistic scenarios, as the power of these representations is to be able to parametrize the data in a continuous manner. The evaluation should then be on the generalization to points not seen by the model. Could the authors elaborate on this point and explain why the metrics that are reported in the section 6 experiments are the right ones?

#### Minor comments, nitpicks and typos

- l.132: Introducing the quantities would help following the paper better for people not familiar with the field. In particular, I guess $x$ is the location in the definition domain $\Omega \in \mathbb R^{d_0}$ and $y$ is the associated scalar value of the signal?
- l.135: super scrit $n$ should be $N$ for the definition of $r$.
- l.139: The notation $X$ and $f(X; \Theta)$ are not defined and kind of confusing. From my understanding, as $K$ should have a size $N\times N$, I guess that $\nabla_\Theta f(X; \Theta)$ is of shape $d \times N$, where $d$ stands for the parameter size?
- l.165: The expression $\mathbb E_f[\dot{\sigma}(f(x)) \dot{\sigma}(f(x'))]$ is not well defined. The expectation over $f$ seems strange, as we don't know where $x, x'$ are coming from. It is unclear what $\dot \sigma$ is. A better explanation of this equation is needed.
- l.166: `grows quadratically with the increase of the number`  -> `grows quadratically with the number`
- l.199: `all $r^t_i$ are linearly transformed by $S^t_{tds}$` -> the linear transform should be $\tilde S_e$ no?
- l.210: The $m$ should be a $n$ as there is $n$ groups.
- l.213: $\tilde S$ should be $\tilde S_e$.
- l.263: The definition of $G$ is not clear, as it should either depend on $i$ or be a min over $i$, with only one term. Could the authors clarify its definition?
- l.269: Giving explicit value or at least a link to where $v, k, R_0$ are defined would help the reader.
- l.286: `the dynamics of f (xi , Θ) at time step t` -> does this mean the dynamics following the original NTK or the dynamics following the empirical NTK?
- l.288: It is not clear why the precision about $\tilde K_e(j_i, j)$ is important?
- l.291: `this error decreases as the similarity of data increases` -> This is not clear where this comes from in the theorem.
- l.893: `, that a variant` -> `that is a variant`?

---

### Note · Authors · 2024-11-13

I have read and agree with the venue's withdrawal policy on behalf of myself and my co-authors.